# Indisulam targets RNA splicing and metabolism to serve as a therapeutic strategy for high-risk neuroblastoma

Anke Nijhuis[1,6], Arti Sikka[1,6], Orli Yogev[2,6], Lili Herendi[1], Cristina Balcells [1], Yurui Ma[1], Evon Poon [2], Clare Eckold[1], Gabriel N. Valbuena [1], Yuewei Xu [1], Yusong Liu[1], Barbara Martins da Costa[2], Michael Gruet [1], Chiharu Wickremesinghe [1], Adrian Benito[1], Holger Kramer [3], Alex Montoya[3], David Carling [3], Elizabeth J. Want[4], Yann Jamin [5], Louis Chesler [2] & Hector C. Keun [1,4✉]

Neuroblastoma is the most common paediatric solid tumour and prognosis remains poor for high-risk cases despite the use of multimodal treatment. Analysis of public drug sensitivity data showed neuroblastoma lines to be sensitive to indisulam, a molecular glue that selectively targets RNA splicing factor RBM39 for proteosomal degradation via DCAF15-E3-ubiquitin ligase. In neuroblastoma models, indisulam induces rapid loss of RBM39, accumulation of splicing errors and growth inhibition in a DCAF15-dependent manner. Integrative analysis of RNAseq and proteomics data highlight a distinct disruption to cell cycle and metabolism. Metabolic profiling demonstrates metabolome perturbations and mitochondrial dysfunction resulting from indisulam. Complete tumour regression without relapse was observed in both xenograft and the Th-*MYCN* transgenic model of neuroblastoma after indisulam treatment, with RBM39 loss, RNA splicing and metabolic changes confirmed in vivo. Our data show that dual-targeting of metabolism and RNA splicing with anticancer indisulam is a promising therapeutic approach for high-risk neuroblastoma.

[1] Department of Surgery & Cancer, Imperial College London, London, UK. [2] Division of Clinical Studies, The Institute of Cancer Research, London, UK. [3] Medical Research Council London Institute of Medical Science, London, UK. [4] Department of Metabolism, Digestion and Reproduction, Imperial College London, London, UK. [5] Division of Radiotherapy and Imaging, The Institute of Cancer Research, London and Royal Marsden NHS Trust, London, UK. [6] These authors contributed equally: Anke Nijhuis, Arti Sikka, Orli Yogev. ✉email: h.keun@imperial.ac.uk

Neuroblastoma is the most common solid and extra-cranial paediatric tumour, originating from neural crest cells of the sympathetic ganglia. Treatment options include surgical resection, cytotoxic chemotherapy, radiotherapy, myeloablative autologous stem cell transplantation and multimodal therapy. However, the prognosis for high-risk cases remains poor with a high incidence of tumour relapse[1]. Most high-risk cases are characterised by *MYCN*-amplification which accounts for 20% of all neuroblastoma cases[2].

Indisulam (E7070) is one of a class of aryl sulfonamides originally discovered by Eisai through several screens for small molecule inhibitors that block cell cycle progression[3,4]. Thereafter, studies revealed that indisulam targets multiple checkpoints through G1 and G2 phases of the cell cycle, and disturbs and downregulates cyclin A, cyclin B, CDK2 and CDC2 via p21/p53 dependent mechanisms[5]. Tumour regression in HCT116 xenografts by indisulam was superior to other anticancer compounds such as 5-FU and Irinotecan[6], which prompted the investigation of indisulam in Phase I/II clinical trials as an anticancer agent for several advanced solid tumours[7–13]. Despite acceptable toxicity profiles, clinical responses have been modest and the efficacy of indisulam has never been tested in neuroblastoma.

More recently, the precise molecular mechanism of action for indisulam was uncovered by two independent studies[14,15]. Indisulam induces a ternary protein complex between RNA Binding Motif 39 (RBM39) and the E3 ubiquitin ligase receptor DDB1 and CUL4 associated factor 15 (DCAF15) resulting in rapid proteasomal degradation of RBM39, aberrant RNA splicing and cell death (Fig. 1a)[14,15]. DCAF15 expression was shown to be necessary for this mode of action and thus proposed as a stratification marker of response in haematopoietic malignancies[14]. RBM39 is an SR-rich protein homologous to, and associated with, the key splicing factors U2 auxiliary factor 65 (U2AF65)[16–18] and SF3b155[19]. RBM39 has been proposed to serve as a pre-mRNA splicing factor[16,20,21] and loss of RBM39 causes alternative splicing defects[22]. Additionally, RBM39 is known to be a coactivator of transcription factors such as AP-1 and oestrogen receptors (ER) α and β[21] and may also regulate metabolism via activation of the NF-κB/c-Myc pathway[23].

In this study, we demonstrate that indisulam is an extremely effective anticancer agent in models of neuroblastoma. Indisulam reduced cellular growth and induced apoptosis in vitro and caused a complete remission of tumours in two in vivo models of neuroblastoma. Indisulam treatment results in DCAF15-dependent RBM39 degradation leading to splicing errors and reduced levels of proteins involved in cell cycle and metabolism. Metabolome perturbations were observed through liquid chromatography–mass spectrometry (LC–MS) profiling and $^{13}$C isotope tracer experiments with glucose and glutamine, and global changes to serine, glycine one-carbon metabolism was also confirmed in vivo. Intriguingly, we uncover some indisulam-mediated metabolic flux alterations from glutamine that could be DCAF15-independent. Finally, we confirm that *MYCN* is a determinant for response to indisulam in various in vitro models. In conclusion, our findings suggest that MYCN-amplified high-risk neuroblastoma may be particularly sensitive to the selective loss of RBM39 and modulation of splicing and metabolism by indisulam.

## Results

### Indisulam causes growth inhibition, selective depletion of RBM39 and global RNA mis-splicing in cellular models of neuroblastoma.
We first sought to investigate tumour types that are likely to respond to aryl sulfonamides by probing publicly available databases with measurements of indisulam efficacy

across 758 cancer cell lines[24]. When sensitivity (as the area under the curve) was reviewed by tumour type, cell lines of the neuroblastoma lineage showed the greatest sensitivity (Fig. 1b) in comparison with cell lines from other lineages (Fig. 1c). The efficacy of indisulam was confirmed in two in vitro models of neuroblastoma, IMR-32 and KELLY. Indisulam reduced growth and viability in both monolayer culture and in 3-D spheroids (Fig. 1d, e). In addition, increased caspase activity was observed in IMR-32 indicating cell death by apoptosis (Supplementary Fig. 2). To confirm that indisulam induces selective RBM39 degradation in neuroblastoma, we performed LC–MS-based global label-free proteomics following indisulam treatment in IMR-32 cells. After 6 h treatment (5 μM) we observed a highly selective loss of RBM39 abundance (~9-fold reduction) compared to ~4300 other detected proteins (Fig. 2a). The degradation of RBM39 was validated by western blot in IMR-32 and KELLY cell lines (Fig. 2b). The dependency on proteasome function for this response was confirmed through the rescue of RBM39 degradation using the proteasomal inhibitor bortezomib (Supplementary Fig. 3). Since the loss of RBM39 is associated with defects in RNA splicing, we sought to identify transcripts that were affected following indisulam treatment in IMR-32 cells using total RNAseq. Stringent detection of altered events such as the skipping of cassette exons or the incorrect inclusion of introns was performed using *SpliceFisher* (github.com/jiwoongbio/SpliceFisher[14], Supplementary Fig 1). Indisulam caused a high number of significant exon skipping (1893) and intron retention (1571) events (Fig. 2c). These include splicing events consistently reported following indisulam exposure such as the skipping of exon 6 and 7 of *TRIM27* (Fig. 2d—black arrows), and both exon skipping and intron retention in *EZH2* (Supplementary Fig. 4)[14,15]. Using polymerase chain reaction (PCR) assays we validated dose- and time-dependent exon skipping of *TRIM27* (Fig. 2e, f) and *EZH2* (Supplementary Fig. 4) in both KELLY and IMR-32 cell lines. Collectively, these data were consistent with the hypothesis that selective degradation of RBM39 and defects in pre-mRNA splicing are likely to be the primary mechanism for the anti-proliferative effect of indisulam in neuroblastoma.

### Indisulam abrogates proteins involved in cell cycle and metabolism through RBM39-mediated alternative splicing.
To capture the consequence of aberrant alternative splicing on protein levels in an unbiased way, we integrated transcriptomic and proteomic analyses post indisulam treatment in IMR-32 neuroblastoma cell line at a later time point (5 μM, 16 h). RBM39 remained the most down-regulated protein but we also observed the dysregulation of other targets (Fig. 3a). By comparing to mis-splicing events we observed that down-regulated proteins largely overlapped with transcripts that were mis-spliced (Fig. 3b, 231/367 (62%) of downregulated proteins were mis-spliced in contrast to 87/502 (17%) upregulated proteins). Of these 231 transcripts, the majority experienced intron retention (174/231 = 75%), which is in line with the observation that the presence of intron-retained transcripts correlates to reduced protein levels often through nonsense-mediated decay[25]. To gain insight into the pathways affected through this mechanism, the 231 target genes were subjected to enrichment analysis which revealed that pathways associated with cell cycle and metabolism were particularly impacted. (Fig. 3b, Supplementary Data 1). Indisulam was originally discovered as an inhibitor of cell cycle[3,4] which is consistent with the enrichment among down-regulated proteins of those involved in cell cycle progression. For example, the proteomic analysis showed a significant decrease of the cyclin-dependent kinase CDK4, KIF20A and BUB1 in IMR-32 cells (Fig. 3a, red dots), all of which undergo mis-splicing following

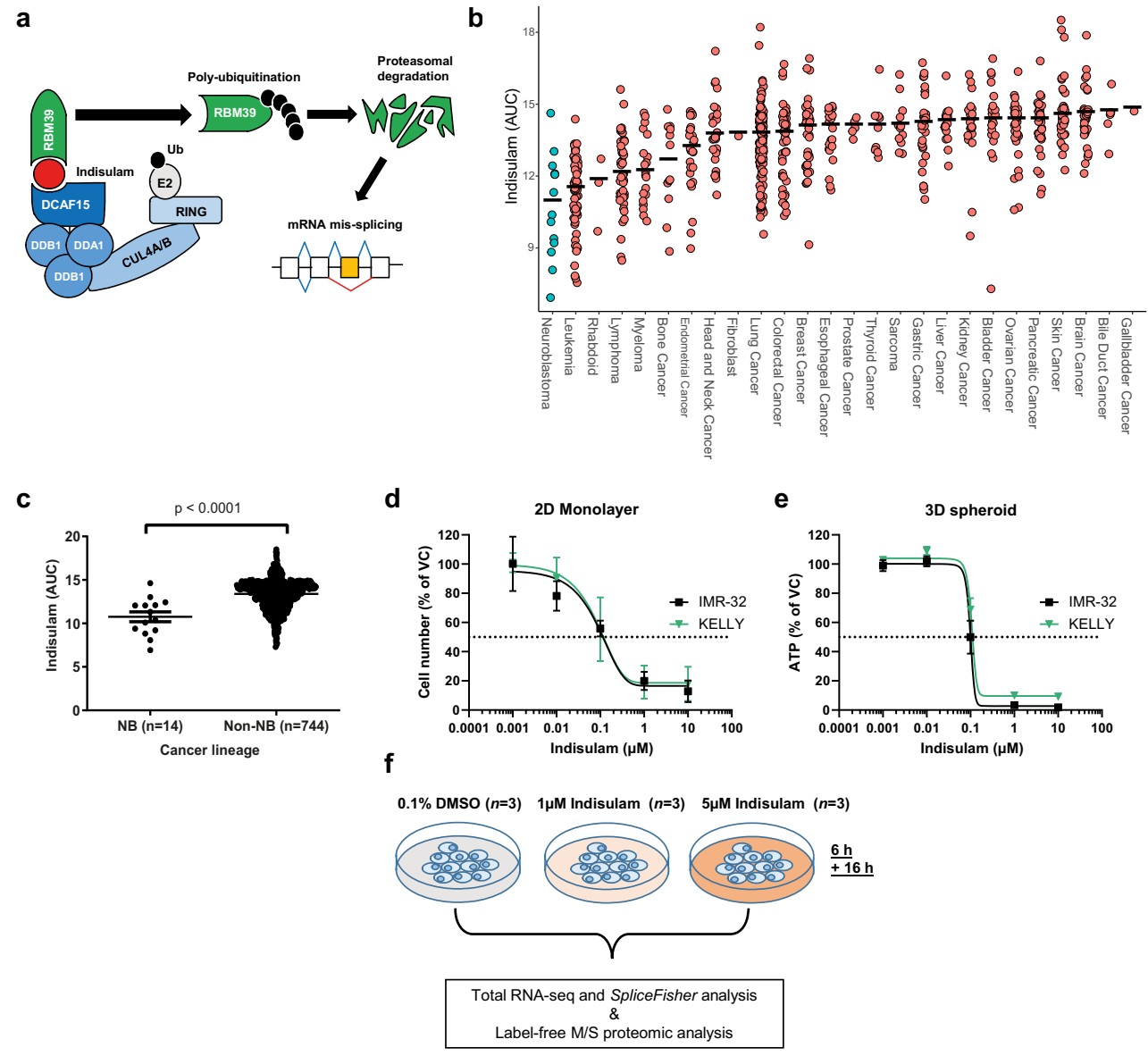

**Fig. 1 Indisulam is highly efficacious in in vitro models of neuroblastoma. a** Aryl sulfonamides such as indisulam (red) act as a molecular glue bringing together DCAF15 E3 ubiquitin ligase and RNA binding protein RBM39 resulting in poly-ubiquitination and degradation of the protein. Depletion of RBM39 leads to aberrant splicing defects. **b**. Median indisulam area-under-curve (AUC) in cell lines from 26 tumour origins. Data was acquired from the CTD[2] network[24], each circle represents one cell line. **c** Indisulam AUC of 14 neural-crest derived neuroblastoma (NB) compared to non-NB cell lines (all other cancer lineages, $n = 744$ cell lines). Each circle represents one cell line. Data were acquired from the CTD[2] network[24] **d** Neuroblastoma lines IMR-32 and KELLY were treated with a 5-point dose-response of indisulam or vehicle control (VC, 0.1% DMSO) for 72 h. Cell growth determined by SRB assay ($n = 3$ independent experiments). **e** IMR-32 and KELLY cells were grown in 3D spheroids and treated with indisulam or VC (0.1% DMSO) for 72 h. ATP was measured with Cell-Titre-Glo 3D. (IMR-32 $n = 3$ independent experiments, KELLY $n = 2$ independent experiments). **f** Overview of workflow of proteomic and RNAseq analysis in IMR-32. Data are represented as mean values ± SD. **d**, **e** Statistical significance in **c** was determined by a Mann Whitney test. Source data is provided as a Source Data File.

indisulam treatment. SpliceFisher analysis showed that *CDK4* undergoes exon skipping of exon 2,3 and 4 and PCR analysis with custom primers confirmed a dose and time-dependent loss of full-length *CDK4* transcript (exon 1–5, 696 bp) in IMR-32 and KELLY cells (Fig. 3d, e). The presence of shorter transcripts where exons have been skipped was also revealed, with concomitant reduction of CDK4 protein levels (Fig. 3f, g). Notably, proteins involved in RNA processing and spliceosome are amongst the most upregulated (Fig. 3a, yellow dots), suggesting cells may be activating compensatory mechanisms following the loss of RBM39.

The observation that one-carbon metabolism and lipid pathways are affected by RBM39 loss has not been reported previously. For example, intron retention in thymidylate synthase (*TYMS*) between exon 4 and 5 was observed (RNAseq read counts, IGV, Fig. 3c). Western blot analysis showed complete depletion of TYMS signal which could be due to loss of the n-terminal epitope of the antibody. To confirm that these events were associated with RBM39 loss and not lineage-specific, we conducted knockdown of RBM39 via siRNA in HCT116 cells and confirmed the reduction in CDK4 and TYMS and mis-splicing events (Supplementary Fig. 5). Together, these data demonstrate

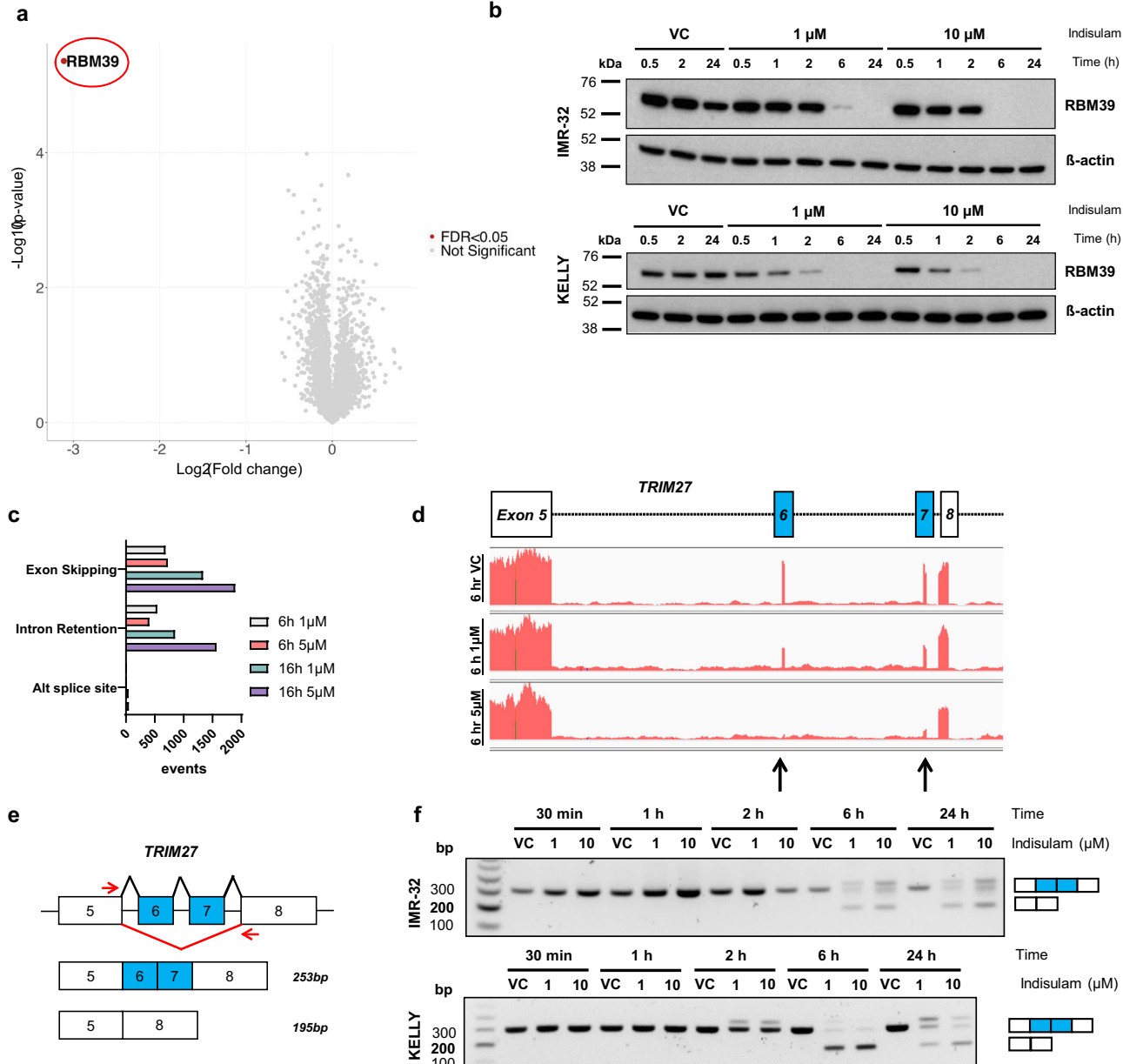

**Fig. 2 Indisulam causes selective degradation of RBM39 and subsequent mis-splicing of RNA. a** Volcano plot of proteomic analysis in IMR-32 cells treated with 5 μM indisulam or VC (0.1% DMSO) for 6 h. RBM39 is highlighted in red. **b** Western blot of IMR-32 and KELLY cells treated with vehicle control (VC), 1 or 10 μM indisulam for 0.5–24 h. Representative blot for n = 2 independent experiments. **c** Number of RNA splicing events (SpliceFisher analysis, n = 3 independent experiments) in IMR-32 cells following treatment with VC or 5 μM indisulam for 6 or 16 h. **d** RNA read counts of *TRIM27* (exon 5–8) in IMR-32 cells treated with VC (top), 1 μM (middle), or 5 μM (bottom) indisulam for 6 h. Black arrows indicate loss of exons 6 and 7. Plot generated with IGV. **e** Diagram of custom primers to detect skipping of exon 6 and 7 of *TRIM27*. **f** PCR of *TRIM27* (exon 5–8) in IMR-32 (top) and KELLY (bottom) cells treated with VC, 1 or 10 μM indisulam for 0.5–24 h. Representative blot for n = 2 independent experiments. Source data is provided as a Source Data File.

that aberrant protein levels in the cell cycle and metabolic pathways are a direct consequence of the loss of RBM39 and erroneous splicing of RNA following indisulam treatment and are not lineage-specific responses to indisulam.

**DCAF15 expression is necessary for the degradation of RBM39 and downstream mis-splicing.** The expression of DCAF15 E3 ligase has been suggested to be critical for the mode of action of aryl sulfonamides including indisulam[14,15]. To confirm that this was the case, we tested the correlation between gene expression and indisulam sensitivity using a large compound sensitivity data

and gene expression for 758 cancer cell lines (The Cancer Target Discovery and Development Network[24]). Across all cell lines, sensitivity to indisulam was significantly correlated to *DCAF15* mRNA expression (Supplementary Fig. 6a). In addition, gene expression from over 1100 cell lines in the cancer cell line encyclopaedia database showed that levels of *DCAF15* in neuroblastoma cell lines was the highest among all solid tumour types (Supplementary Fig. 6b), further supporting the hypothesis that neuroblastoma patients could represent a target population for therapeutic intervention with aryl sulfonamides, such as indisulam.

To study the essentiality of DCAF15 in the mode of action of aryl sulfonamides in neuroblastoma, we generated DCAF15

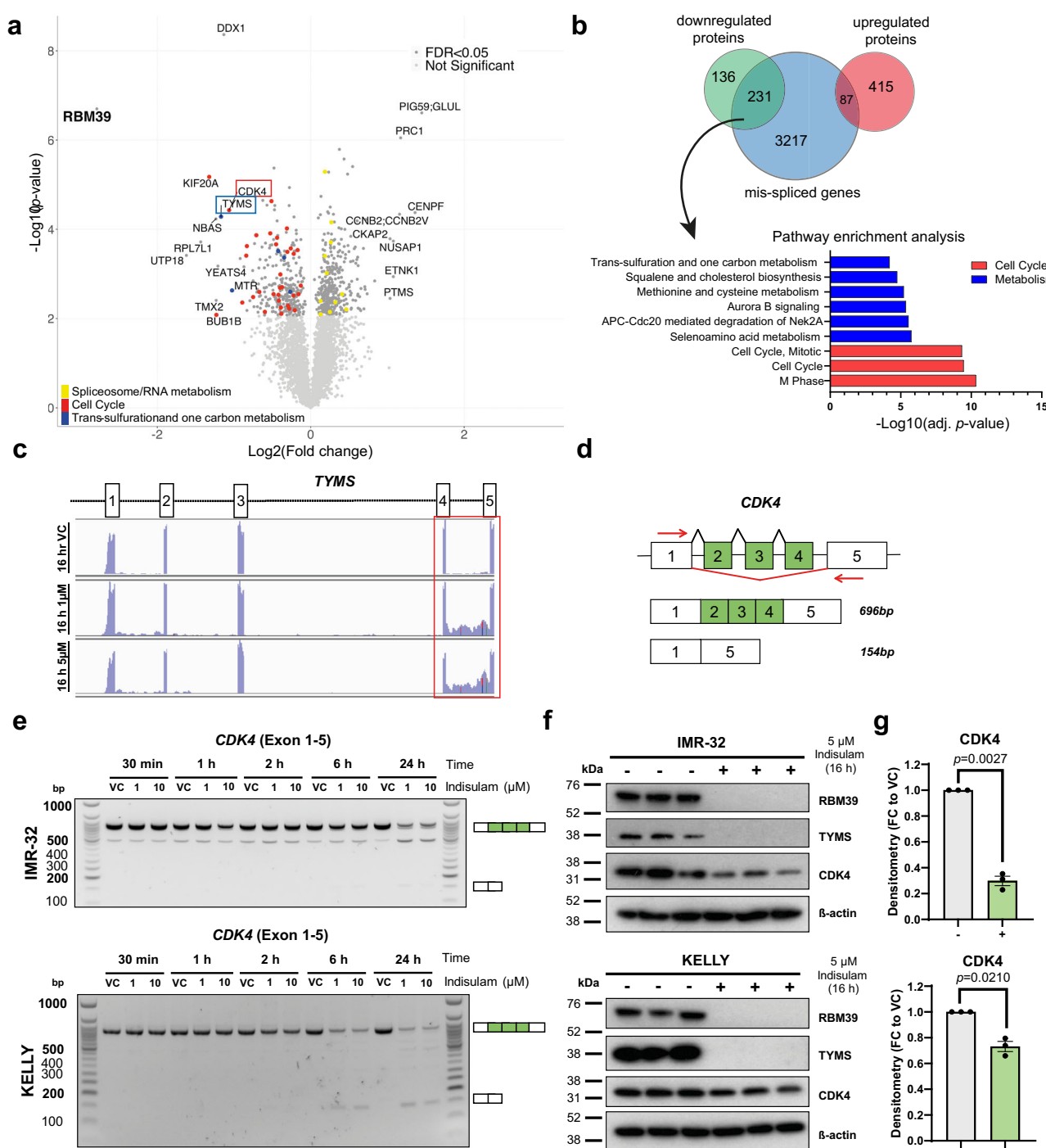

**Fig. 3 Indisulam-mediated RBM39 degradation leads to the mis-splicing and depletion of proteins regulating cell cycle and metabolism. a** Volcano plot of proteomic analysis in IMR-32 cells treated with 5 μM indisulam or Vehicle control (VC, 0.1% DMSO) for 16 h. **b** Overlap of splicing events (exon skipping and intron retention) with up or down-regulated proteins after 16 h of Indisulam. Gene ontology analysis of mis-spliced down-regulated proteins and pathway analysis. **c** RNA Read counts of *TYMS* (exon 1–5) in IMR-32 cells treated with VC (top), 1 μM (middle) or 5 μM (bottom) indisulam for 16 h. The red box indicates intron retention between exon 4 and 5. **d** Diagram of custom primers to detect skipping of exon 2–4 of *CDK4*. **e** PCR of *CDK4* (exon 1–5) in IMR-32 (top) and KELLY (bottom) cells treated with VC, 1 μM or 10 μM indisulam for 0.5–24 h. Representative blot for *n* = 2 independent experiments. **f** Western blot analysis of IMR-32 (top) and KELLY (bottom) following treatment of VC or 5 μM indisulam for 16 h. Membranes probed for RBM39, TYMS, CDK4 and ß-actin (*n* = 3 independent experiments). **g** Densitometry analysis of CDK4 in IMR-32 (top) and KELLY (bottom). Data are presented as mean values ± SEM. Statistical significance was determined by a two-tailed one-sample *t*-test. Source data is provided as a Source Data File.

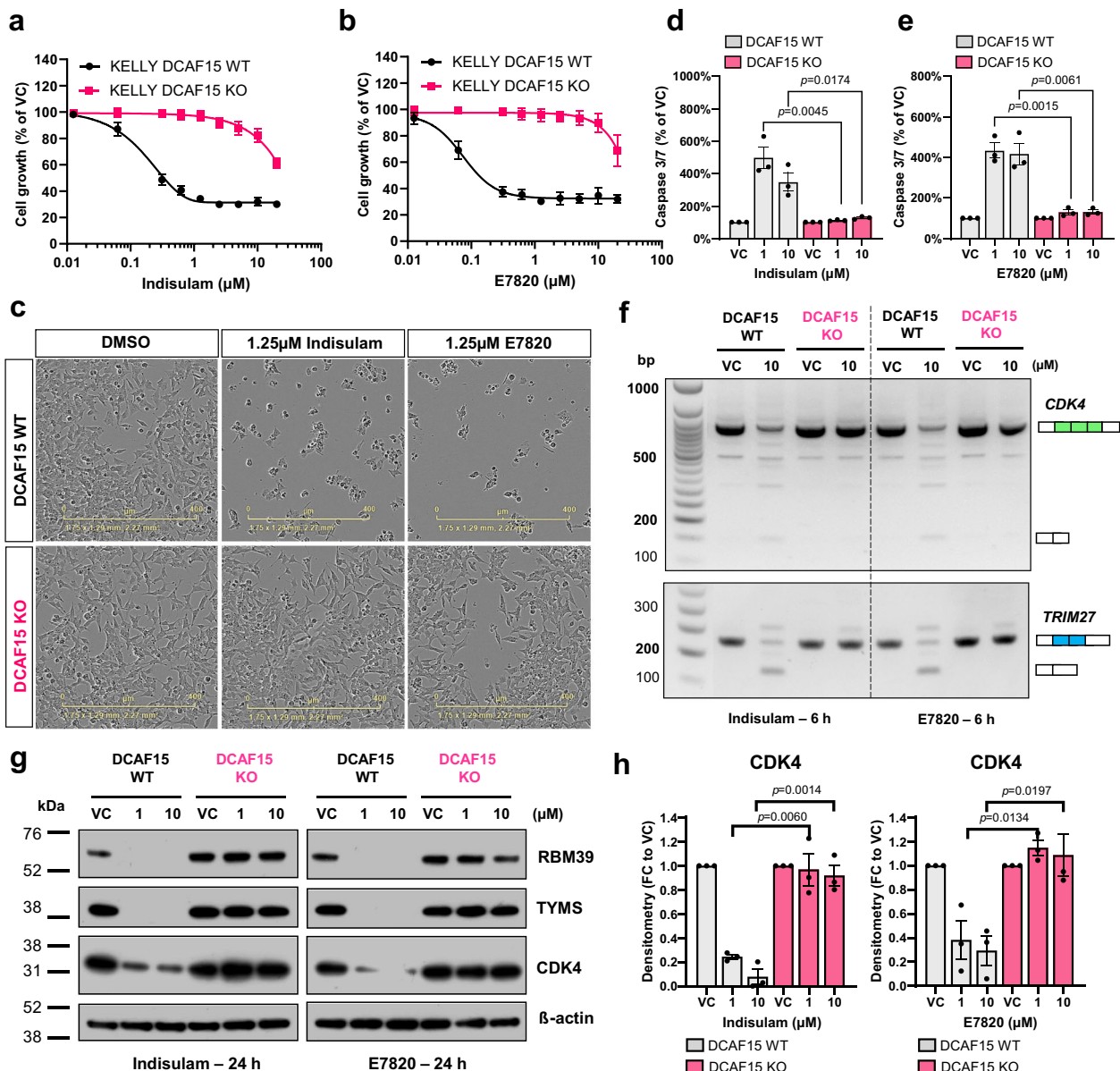

**Fig. 4 RBM39 degradation and RNA mis-splicing via aryl sulfonamides is DCAF15 dependent. a, b** KELLY DCAF15[WT] or DCAF15[KO] cell line treated with aryl sulfonamide indisulam (**a**) or E7820 (**b**) for 72 h. Cell growth was measured by SRB assay. $n = 3$ independent experiments. **c** Representative images of KELLY DCAF15[WT] or DCAF15[KO] cells treated with 1.25 μM indisulam or E7820 (IncuCyte, Sartorius) ($n = 3$ independent experiments). The scale bar is 400 μm. **d, e** Caspase 3/7 signal following 48 h treatment of indisulam (**d**) or E7820 (**e**). Data normalised to cell mass (SRB) ($n = 3$ independent experiments). **f** PCR gel depicting exon skipping of *CDK4* (exon 2–4) and *TRIM27* (exon 6–7) of KELLY DCAF15[WT] or DCAF15[KO] cells treated with indisulam or E7820. Representative image of $n = 3$ independent experiments. **g** Western blot analysis of RBM39, TYMS, CDK4 and β-actin following 24 h treatment of indisulam or E7820 (representative blot of $n = 3$ independent experiments). **h** Densitometry of CDK4 protein analysis of (**g**). Data are presented as mean values ± SD (**a, b**) or SEM (**d, e, h**). Statistical significance in (**d, e, h**) was determined by a two-tailed unpaired *t*-test. Source data are provided as a Source Data File.

knockout cells using CRISPR–Cas9 in KELLY (Supplementary Fig. 7) in which loss of DCAF15 rescued acute RBM39 degradation from indisulam treatment (6 h, 10 μM, Supplementary Fig. 8). Growth inhibition and apoptosis induction by two aryl sulfonamides indisulam and E7820 were rescued in DCAF15[KO] clones (Fig. 4a–e). Further, indisulam and E7820 induced exon skipping in both *CDK4* and *TRIM27*, observable in DCAF15[WT], and not in DCAF15[KO] cells (Fig. 4f). Experiments also confirmed that both aryl sulfonamides reduced CDK4 and TYMS protein levels in a DCAF15-dependent manner (Fig. 4g, h). This mirrored observations in IMR-32 and KELLY cells with siRNA knockdown of DCAF15 (Supplementary Fig. 9). Thus, it

was confirmed that DCAF15 is necessary for the mechanism of action of two RBM39-degraders indisulam and E7820 in neuroblastoma including downstream consequences such as RNA mis-splicing and perturbations to key factors in cell cycle progression and metabolism.

**Indisulam targets metabolism in a DCAF15-dependent and independent manner.** Indisulam was previously reported to be a potent inhibitor of the extracellular carbonic anhydrase IX (CAIX) in cell-free assays[26]. Also, RBM39 (or CAPERα) has been demonstrated to be a regulator of mitochondrial respiration and glucose-derived carbon flux into tricarboxylic acid (TCA) cycle intermediates[23].

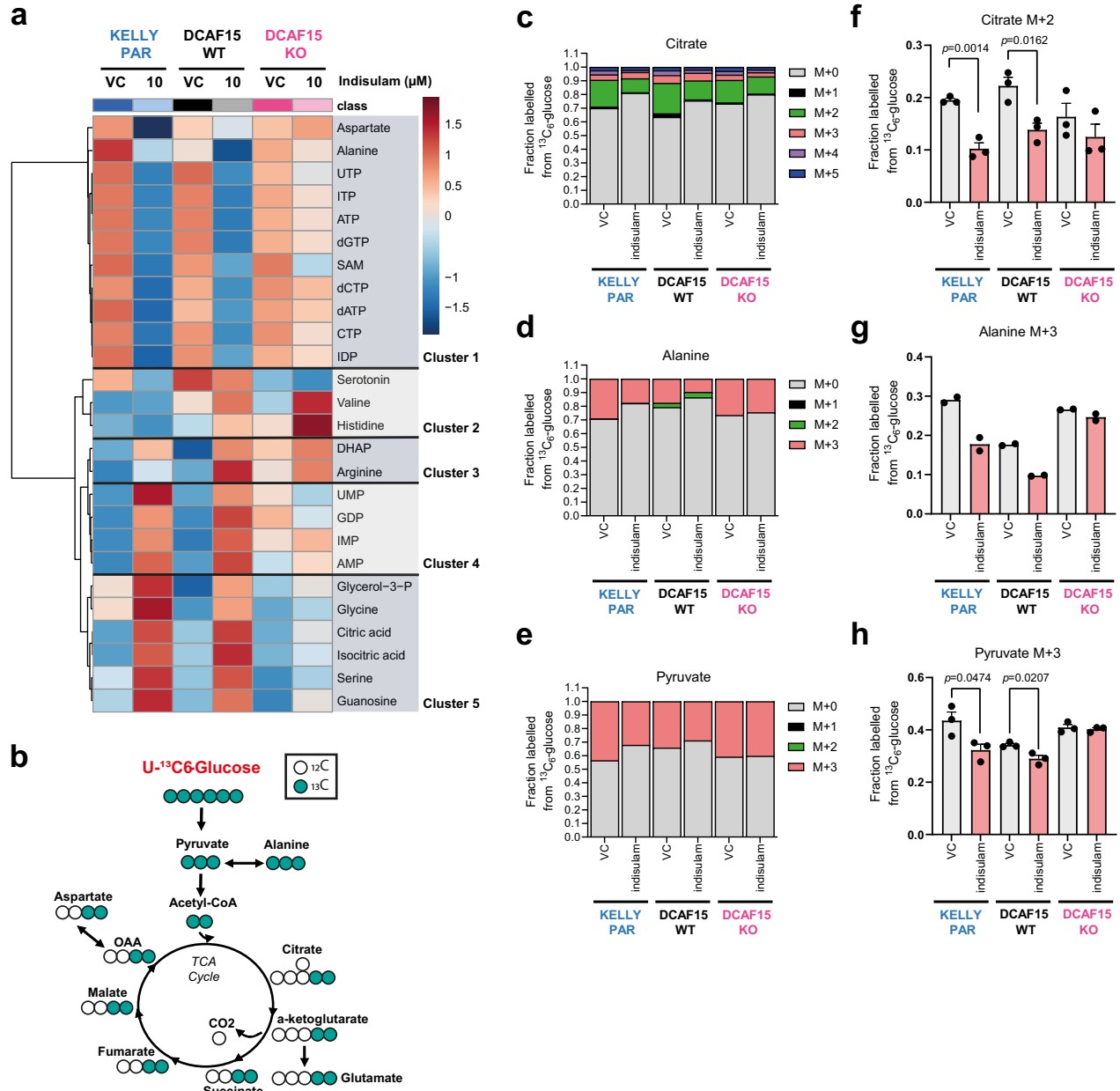

**Fig. 5 Indisulam modulates the metabolome and glucose utilisation. a** KELLY Parental (PAR), DCAF15[WT] or DCAF15[KO] dosed with vehicle control (VC, 0.1% DMSO) or 10 μM indisulam for 24 h and intracellular metabolites analysed by HILIC LC–MS/MS. Heatmap showing hierarchal clustering of significantly altered metabolites ($p$-adj < 0.001, Kruskal–Wallis test, MetaboAnalyst). Data are an average of four technical replicates of $n = 2$ independent experiments. **b** Schematic representation of enrichment of [13]C derived from [13]C6-glucose into TCA cycle intermediates detected by GC-MS. **c–h** KELLY PARENTAL (PAR), DCAF15[WT] or DCAF15[KO] cells were exposed to indisulam in the presence of 5.6 mM [13]C6-glucose for 24 h. Fraction of carbons labelled from [13]C6-Glucose in Citrate (**c**), Alanine (**d**) and Pyruvate (**e**). M + $n$: a metabolite with n carbon atoms labelled with [13]C. Data are mean of $n = 3$ independent experiments, $n = 2$ for Alanine. Changes in M + 2 labelled Citrate (**f**), M + 3 labelled Alanine (**g**) and M + 3 labelled Pyruvate (**h**) following indisulam exposure. ($n = 3$ independent experiments, $n = 2$ for Alanine). Data are presented as mean values ± SEM (**f–h**). Statistical significance in **f** and **h** was determined by a two-sided unpaired $t$-test. Source data is provided as a Source Data File. UTP uridine triphosphate, ITP inosine triphosphate, ATP adenosine triphosphate, dGTP deoxyguanosine triphosphate, SAM S-Adenosyl methionine, dCTP deoxycytidine triphosphate, dATP deoxyadenosine triphosphate, CTP cytidine triphosphate, IDP inosine diphosphate, DHAP dihydroxyacetone phosphate, UMP uridine monophosphate, GDP guanosine diphosphate, IMP inosine monophosphate, AMP adenosine monophosphate.

Therefore, we explored further the impact of indisulam on cellular metabolism in KELLY DCAF15[WT] and DCAF15[KO] using metabolomics and [13]C-based metabolic flux analysis.

Both metabolic profiling and stable isotope tracing experiments show that indisulam induces a global metabolic change in DCAF15[WT] cells, disrupting energy metabolism and affecting the metabolic fate of both glucose and glutamine. After controlling for false discovery, 26 metabolites exhibited significant differences in abundance across conditions ($p_{adj} < 0.01$, Kruskal–Wallis test) and hierarchical clustering highlighted the presence of up to 5 distinct clusters (Fig. 5a, Supplementary Fig. 11, Supplementary Data 3). In line with the cellular toxicity observed, indisulam

lowered levels of ATP and other nucleotide triphosphates (cluster 1), with concomitant increases in levels of AMP and other nucleotide monophosphates (cluster 4) in DCAF15$^{WT}$; these effects were rescued in DCAF15$^{KO}$ cells. (Fig. 5a). Cluster 5 defined additional metabolites increasing with indisulam treatment in DCAF15$^{WT}$, including the TCA cycle intermediates citrate and isocitrate, as well as the amino acids serine and glycine. The increase in serine and glycine following indisulam treatment was also observed in IMR-32 cells by two mass-spectrometry methods (Supplementary Fig. 10), confirming this modulation occurred consistently across neuroblastoma models. Serine and glycine are intermediates in the one-carbon metabolic pathway, identified as modulated by indisulam in our RNAseq and proteomics data (Fig. 3a, b). The treatment-related response in cluster 5 metabolites was rescued in DCAF15$^{KO}$ cells, although the reduction of the effect varied across metabolites. Interestingly the most clearly DCAF15-independent metabolite, S-adenosyl methionine, is also a key component on the one-carbon pathway and is the universal methyl donor for histone and DNA methylation. Thus, it is possible that indisulam may cause DCAF15-independent perturbations to one-carbon metabolism that could impact epigenetic regulation and potentially contribute to off-target effects of indisulam.

Incubation with U-$^{13}$C-glucose showed a significant decrease in label incorporation into [M + 3] pyruvate, [M + 3] alanine and [M + 2] citrate upon treatment (Fig. 5c–h) in DCAF15$^{WT}$ cells, denoting a decrease in glycolytic flux and glucose oxidation, whereas no significant changes were found in DCAF15$^{KO}$ cells. We hypothesised that the accumulation of citrate (Fig. 5a) despite decreased synthesis, could result from decreased de novo lipogenesis, which relies on citrate exported from mitochondria to synthesise the lipid precursor, cytosolic acetyl-CoA. A decrease in label incorporation of glucose carbon into esterified palmitate was observed (Supplementary Fig. 12, Supplementary Data 2) consistent with this hypothesis. Disruption of acetyl-CoA-dependent sterol biosynthesis upon indisulam treatment was also visible in the proteomics data (Fig. 3a, b) which could further reduce the utilisation of citrate for anabolism.

Next, we sought to explore the fate of glutamine as another potential anaplerotic source for TCA cycle and energy production, using U-$^{13}$C-glutamine as a tracer. Our results indicate a suppression of glutaminolysis by indisulam which was less pronounced than for glycolysis, and, importantly, appears to be DCAF15-independent (Fig. 6b–d), with similar changes in label incorporation from glutamine into citrate, aspartate, and malate (Fig. 6b–g).

Further, indisulam reduced oxygen consumption rate (OCR) in IMR-32 (Supplementary Fig. 13) and in DCAF15$^{WT}$ and DCAF15$^{KO}$ cells, i.e. DCAF15-independent (Fig. 6h–j), suggesting the impact on mitochondrial metabolism and perhaps glutamine oxidation is not dependent on RBM39 degradation. Notably, E7820 did not reduce OCR and thus the impact on respiration is potentially unique to indisulam an off-target effect. In addition, indisulam and E7820 significantly depolarised the mitochondrial membrane (Fig. 6k) confirming an impact on mitochondrial function.

In summary, these data show that indisulam causes significant impact on cellular metabolism including the accumulation of substrates for DNA and lipid synthesis including citrate and one-carbon donors (serine, glycine), consistent with the RBM39-dependent mis-splicing of biosynthetic genes in lipid and one-carbon metabolism. Furthermore, indisulam disrupted oxidative metabolism in a manner potentially independent of RBM39 depletion.

**Indisulam is highly efficacious in two mouse models of neuroblastoma.** Given that indisulam showed profound efficacy in

neuroblastoma lines in vitro, indisulam was tested in an in vivo xenograft model. IMR-32 cells were subcutaneously injected into NCr Foxnnu mice, randomised and were dosed intravenously for 8 days with indisulam (25 mg kg$^{-1}$) or vehicle[6]. Complete tumour regression and a 100% survival rate (Fig. 7a–d) were reported in the treated group compared to the vehicle. Notably, no disease relapse was observed for up to 66 days after cessation of treatment. To study pharmacodynamic (PD) markers and confirm mechanism of action in vivo, xenograft tissue was harvested after 4 days of treatment ($n = 4$ for immunofluorescence, $n = 2$ RNA extraction, $n = 10$ metabolic analysis). This time point was chosen based on tumour sizes observed during the survival study. On day 4, xenograft tumour volume was still similar to starting volume (indisulam arm $n = 5$, 94% ± 48%, Supplementary Fig. 14), and immunofluorescence of xenograft tumours ($n = 2$ vehicle, $n = 2$ indisulam) revealed a reduction of RBM39 in the presence of proliferation marker Ki67 i.e. in context of viable tumour. (Fig. 7e, additional samples Supplementary Fig. 15). Mis-splicing of *CDK4* and *TRIM27* (Fig. 7f, g, $n = 1$ vehicle, $n = 1$ indisulam) was observed at the same timepoint, demonstrating evidence of target engagement and mis-splicing of predicted genes confirming the downstream mode of action of indisulam in vivo. Xenograft tumours were also subjected to metabolic analysis via LC–MS ($n = 5$ vehicle, $n = 5$ indisulam) to validate metabolomic changes in the tumour. A significant increase in multiple amino acids such as methionine, serine and glycine were detected (Fig. 7h, i, Supplementary Data 4) in concordance with our in vitro observations of perturbation to in one-carbon metabolism in KELLY and IMR-32. Thus, we were able to demonstrate key PD responses to indisulam in vivo that provide proof-of-concept for target modulation and the proposed downstream mechanism of gene mis-splicing, protein depletion and altered metabolism in neuroblastoma models.

A second murine model was employed to establish further the efficacy of indisulam in vivo; a widely used Th-*MYCN* transgenic engineered mouse model (129×1/SvJ-Tg(Th-*MYCN*)41Waw/Nci). These tumours closely recapitulate many of the clinical and genomic features of high-risk neuroblastoma disease[27,28]. We observed a near complete reduction in tumour volume by MR imaging after 7 days of indisulam treatment compared to vehicle control (Fig. 8a-c). Importantly, no tumour relapse was reported over a period of more than 124 days ($n = 5$) exemplifying the absence of recurrence (Fig. 8b). One animal in the treated group died on day 7, where the autopsy revealed no evidence of residual tumour but a bowel obstruction. Together these data indicate that indisulam may be a highly efficacious therapeutic agent for high-risk neuroblastoma.

**N-Myc is a determinant for sensitivity to aryl sulfonamide treatment.** Amplification of the transcription factor *MYCN*, which along with high c-Myc expression is frequently observed in high-risk neuroblastoma, leads to an increased oncogenic transcriptional programme[29] whereby cells may rely more heavily on rapid and correct pre-mRNA processing and alternative splicing[30]. We, therefore, explored the efficacy of indisulam and E7820 in three *MYCN* amplified (BE2C, IMR5 and KELLY) and three lines without *MYCN*amp (SKNAS, SHEP and SKNSH). Cells with *MYCN* amplification were overall more sensitive to both indisulam and E7820 compared to cells without *MYCN* amplification (Fig. 9a–c). We also used the neuroblastoma Tet21 model in which *MYCN* expression is doxycycline-inducible (Tet-off). Tet21 with *MYCN* expression (*MYCN* ON) were considerably more sensitive to indisulam compared to *MYCN* OFF cells (Fig. 9d). In conclusion, neuroblastoma cells with high *MYCN* levels were more sensitive to both aryl sulfonamides

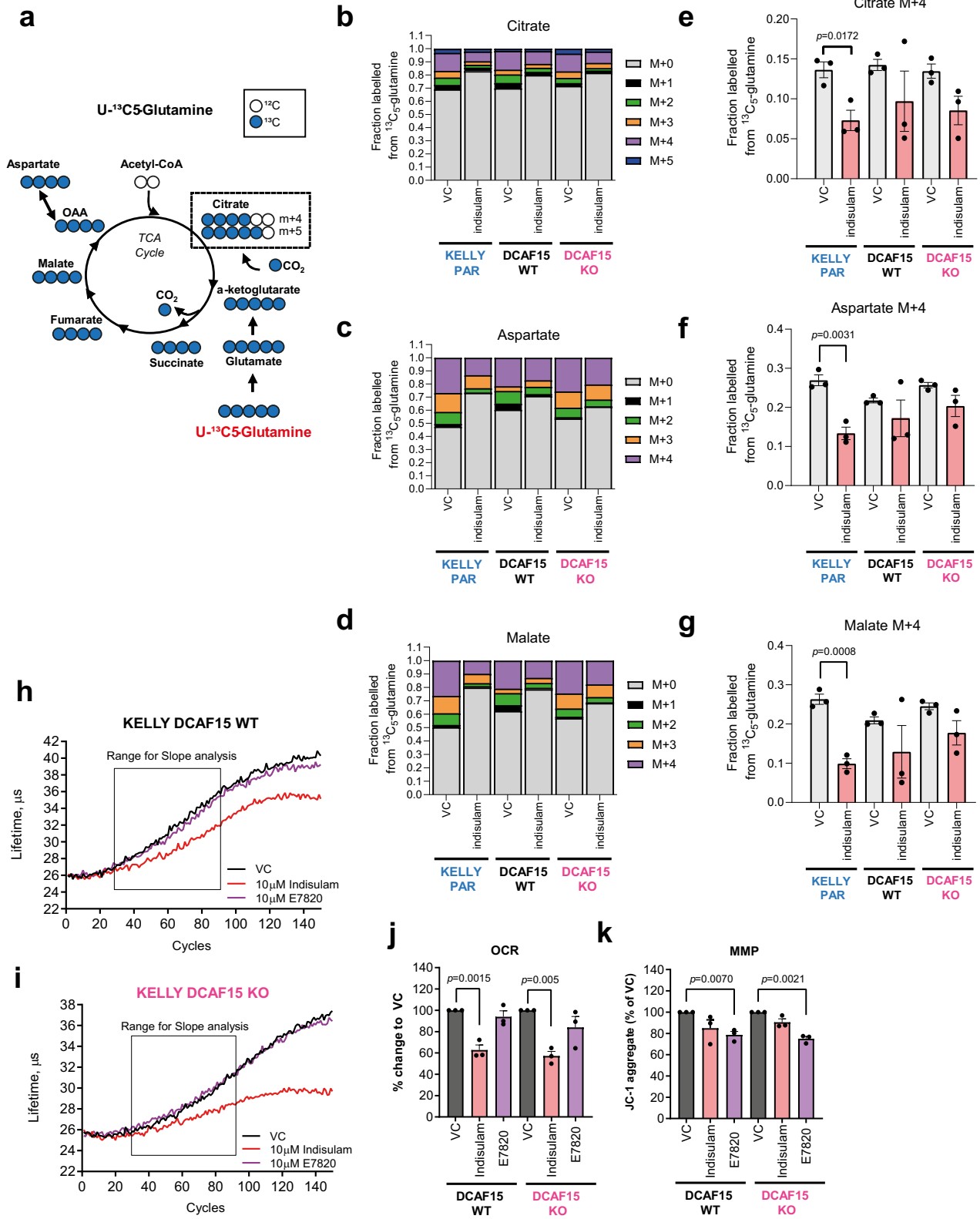

(Fig. 9e) suggesting that this oncogenic subtype of tumours may respond well to these compounds in a clinical setting.

## Discussion

Until recently the key molecular event determining the anti-tumour activity of aryl sulfonamides, selective RBM39 degradation, was unknown and predictive biomarkers for therapy response unavailable. Since that discovery, a number of studies have detailed the structural basis of interactions between aryl sulfonamides, DCAF15 and RBM39, creating both opportunities for the generation of novel protein degrading therapeutics based on DCAF15 targeting as well as new interest in RBM39 as an emerging anticancer target.

**Fig. 6 Indisulam modulates glutamine utilisation and mitochondrial oxidation. a** Schematic representation of enrichment of $^{13}C$ derived from $^{13}C_5$-glutamine into TCA cycle intermediates detected by GC-MS. **b–g** KELLY PARENTAL (PAR), DCAF15$^{WT}$ or DCAF15$^{KO}$ cells were exposed to 10 μM indisulam in the presence of 2 mM $^{13}C_5$-glutamine for 24 h. Fraction of carbons labelled from $^{13}C_5$-glutamine in Citrate (**b**), Aspartate (**c**) and Malate (**d**). M + n: a metabolite with n carbon atoms labelled with $^{13}C$. Data are the mean of three independent experiments. Changes in M + 4 labelled Citrate (**e**), Aspartate (**f**) and Malate (**g**) following indisulam exposure (n = 3 independent experiments). **h–j** Lifetime measurements of MX-Xtra probe in KELLY DCAF15$^{WT}$ (**h**) and DCAF15$^{KO}$ (**i**) treated with VC (0.1% DMSO), 10 μM indisulam or 10 μM E7820. Representative data for n = 3 independent experiments. **j** Oxygen Consumption Rate (OCR) measures in a lifetime (μs) per h. Data are normalised to VC treatment (n = 3 independent experiments). **k** Mitochondrial Membrane Potential (MMP) of KELLY DCAF15$^{WT}$ or DCAF15$^{KO}$ cells treated with VC (0.1% DMSO), 10 μM indisulam or 10 μM E7820. Measured as JC-1 aggregates, normalised to VC. Data are presented as mean values ± SEM. Statistical analysis in **e–g** was determined by a two-sided unpaired t-test, and in **j** and **k** by a two-sided one-sample t-test. Source data is provided as a Source Data File.

To the best of our knowledge, of the 47 clinical trials[18] that have used indisulam or a related aryl sulfonamide, none have tested the efficacy of these compounds in patients with neuroblastoma. This is despite apparent evidence from public datasets demonstrating that neuroblastoma cell lines, together with several lymphomas and leukaemia models, exhibit high susceptibility to indisulam treatment[24]. While recent studies in primary AML cells have confirmed this responsiveness[31] additional pre-clinical evidence for treatment of neuroblastoma was lacking.

Here, we have shown that aryl sulfonamides are a viable and potent therapeutic strategy using multiple in vivo and in vitro neuroblastoma models including a transgenic mouse model of MYCN-driven, high-risk disease. Significantly, indisulam has been assessed in a wide range of xenograft models including SW[32], LX-1, PC9, HCT15, HCT116[6] and MML-AF9 leukaemia cells[31], none modelled neuroblastoma and all previous experiments showed relapse after initial tumour regression once treatment was withdrawn. In contrast, we observe complete tumour response with prolonged remission after cessation of therapy, strongly underlining the fact that high-risk neuroblastoma may be particularly sensitive to indisulam.

Another aspect of our findings is the recognition that aryl sulfonamides may cause broad metabolic deficiencies and growth inhibition in cancer cells via the mechanism of RBM39 depletion and mis-splicing of specific genes which regulate metabolism and cell cycle. Although indisulam-induced depletion of RBM39 was previously reported to cause widespread RNA mis-splicing, these events are far from random. RBM39 knockdown coupled to CLIP-seq has indicated that RBM39 regulates a distinct set of splicing events from that of closely related spliceosomal factors U2AF65 and PUF60, enriched in cell cycle, RNA processing and metabolic pathway genes[22]. Specifically, we demonstrated that loss of RBM39 by either pharmacological or gene silencing is consistently associated with RNA splicing errors and reduced protein expression of critical factors for proliferation such as CDK4 and TYMS across different cell backgrounds. Targeted interference against CDK4/6 has been previously shown to cause G1 arrest in neuroblastoma[33], in part phenocopying indisulam. Also, we have recently shown that targeting cell cycle proteins (such as CDK2/9) is an increasingly important strategy for therapeutic developments in neuroblastoma[34]. As the serine, glycine and one-carbon metabolic pathway is closely linked to pyrimidine metabolism it is plausible that mis-splicing of TYMS activity could explain these observations, although CDK4 loss itself can reduce TYMS expression in a cell cycle independent manner[35]. Importantly, we confirm that these specific PD events translate to the in vivo setting, paving the way for their use as biomarkers for early evaluation of therapeutic response in clinics.

Although DCAF15 is a strong determinant for indisulam sensitivity as shown by us and others[14,15], a recent study suggested that DCAF15 mRNA levels in primary samples of tumour cells of AML patients did not correlate to RBM39 degradation[36]. This highlights the need for additional predictive and PD

biomarkers which we propose in the current work. We also observed some metabolic changes that are not completely rescued in DCAF15$^{KO}$ cells, i.e. independent of RBM39 degradation, which is in accordance with the lack of complete cytotoxic rescue in DCAF15$^{KO}$ cells. Intriguingly, sensitivity to indisulam in neuroblastoma lines is greater than might be predicted by direct siRNA targeting of RBM39 (Supplementary Fig. 16). These on- and off-target effects of indisulam combined might contribute to the particular potency against neuroblastoma.

It has been reported that N-Myc directly regulates an alternative splicing programme by mediating splicing factor expression in high-risk neuroblastoma[37]. Here, we show MYCN amplification as a determinant for aryl sulfonamide sensitivity. As MYCN amplified tumours are more likely to be sensitive to dual CDK4/6 inhibition[33] it is plausible therefore that these may also be more sensitive to CDK4 downregulation downstream of indisulam exposure and RBM39 depletion. Furthermore, MYCN amplification reprogrammes metabolism in neuroblastoma in several ways, including effects to pathways targeted by indisulam by both RBM39-dependent and independent means, such as glutaminolysis[38,39], lipogenesis, the serine-glycine synthesis and one-carbon metabolic pathway[40] and reactive oxygen species production[38]. Combined, the increased dependency of neuroblastoma cells on metabolic reprogramming and RNA processing as a result of high N-Myc/c-Myc activity might be important factors that explain the sensitivity of high-risk neuroblastoma to indisulam.

Altogether, this study demonstrates high-risk neuroblastoma may be particularly sensitive to the dual targeting of metabolism and RNA splicing by the aryl sulfonamide indisulam. Since clinical PK data already exist and because treatment is well-tolerated in many patients, indisulam is therapeutic that could be rapidly repurposed for clinical trials in high-risk neuroblastoma patients. Molecular profiling has revealed several specific gene and pathway level effects that could explain the high vulnerability of neuroblastoma to indisulam. The gene, protein and metabolite biomarkers identified in this study, in addition to DCAF15, provide a solid basis for confirming target engagement, monitoring early response to treatment and, in the future, for personalised therapy. Further studies on the essentiality of RBM39 and how cells may compensate for RBM39 loss could further optimise therapeutic targeting of RBM39 and accelerate biomarker-driven clinical trials.

## Methods

**Cell culture.** IMR-32, BE(2)C and SK-N-AS cells cell were purchased from the American Type Culture Collection (ATCC) (Manassas, VA). KELLY and SH-SY-5Y were obtained from the German Collection of Microorganisms and Cell Culture (DSMZ, Germany). SK-N-SH was purchased from the Human Protein Atlas project (HPA). IMR-5 and Tet21 were a kind gift from Eilers lab, University of Wurzburg and SHEP cells were a kind gift from Weiss lab, University of California at San Francisco. Most cell lines were maintained in DMEM supplemented with 10% foetal bovine serum (FBS), apart from KELLY and IMR-32 were cultured in ATCC's Eagle's Minimum Essential Medium (EMEM) with 10% FBS and 1%

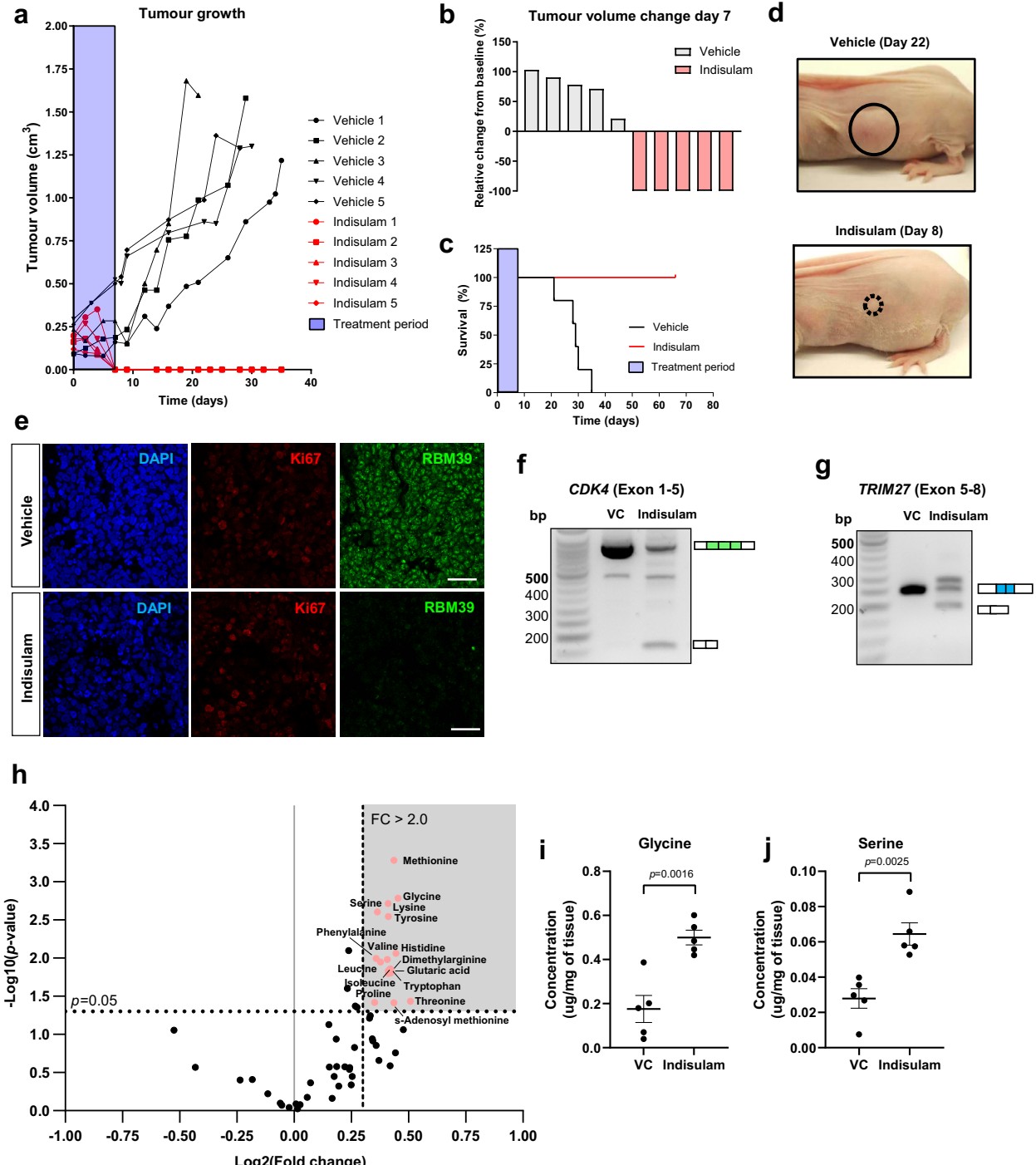

**Fig. 7 Indisulam is efficacious in IMR-32 neuroblastoma in vivo xenografts. a** Tumour growth in mice bearing IMR-32 xenografts treated with either vehicle (*n* = 5) or indisulam (*n* = 5) for 8 days. Tumour volume measured every 2–3 days until the humane end-point was reached. **b** Waterfall plot showing relative changes in tumour volume of xenografts at Day 7. **c** Survival plot of mice bearing IMR-32 xenografts. **d** Representative images of IMR-32 xenografts treated with vehicle (Day 22) or indisulam (Day 8). **e** Immunofluorescent staining of RBM39 and Ki67 in IMR-32 xenograft bearing mice treated with vehicle or indisulam after 4 days. The scale bar is 20 µm **f**. PCR of *CDK4* in IMR-32 xenograft tumour treated with vehicle or indisulam. **g**. PCR of *TRIM27* in IMR-32 xenograft tumour treated with vehicle or indisulam. **h**. Volcano plot of metabolites in IMR-32 xenograft tissues detected by LC-MS (HILIC method). **i**. Glycine and serine changes were detected in xenograft tumours (*n* = 5 samples per group). Data are presented as mean ± SEM. Statistical analysis in **i** and **j** was determined by a two-tailed unpaired *t*-test. Source data is provided as a Source Data File.

penicillin/streptomycin, and IMR5 cells were cultured in RPMI-1640 with 10% FBS. Cell lines were authenticated using short tandem repeat DNA profiling by Public Health England and mycoplasma was tested regularly.

**siRNA knockdown**. IMR-32 or KELLY cells were seeded overnight in 96-well plates. Cells were transfected with SMARTpool DCAF15 siRNA (L-0.31237-01,

Dharmacon, GE Healthcare) or a non-targeting control siRNA (NTC, D-001810-01-05) at a final concentration of 30 nM and Lipofectamine 2000 transfection reagent (Invitrogen, UK). Six-well plates were treated with indisulam (vehicle control, 1 µM or 10 µM indisulam) 48 h post-transfection for 6 h and RNA/protein harvested accordingly. For RBM39 knockdown, HCT116 cells were transfected with SMARTpool RBM39 (L-011965-00, Dharmacon) or NTC at 30 nM. Post-

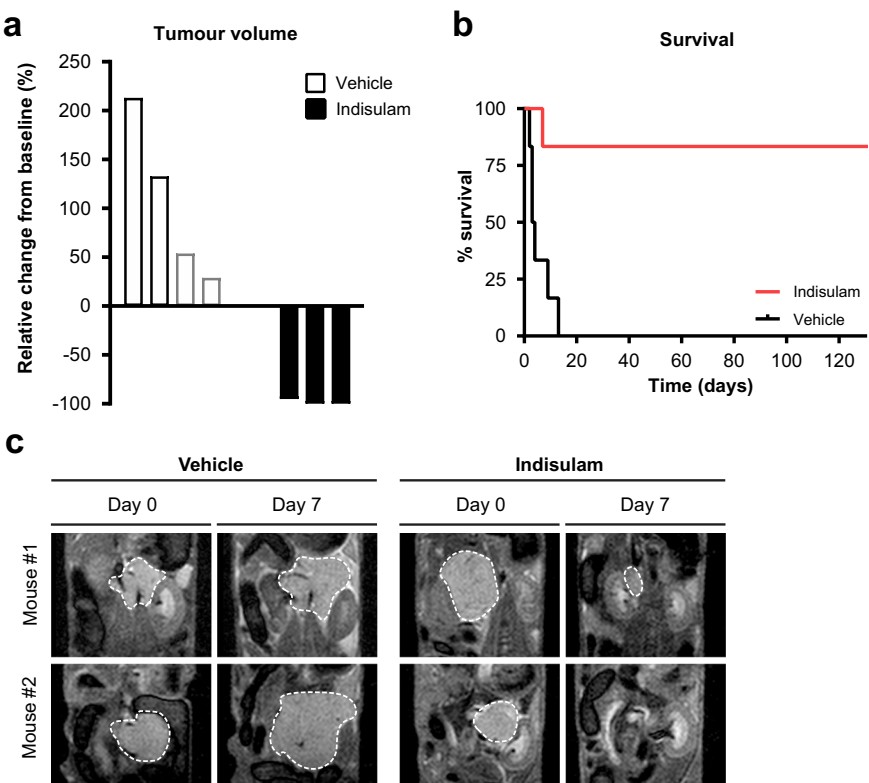

**Fig. 8 Indisulam is efficacious in the Th-*MYCN* transgenic mouse model of neuroblastoma. a** Waterfall plot showing the relative changes in tumour volume in the Th-*MYCN* transgenic mice treated for 7 days with indisulam ($n = 3$ mice) or vehicle control ($n = 4$ mice) measured by MRI. Note that the mice represented by the grey bars reached the tumour size study endpoint (palpation) before their Day 7 MRI scan and the data shown for these two mice are the relative changes in volume following day 3 of treatment with indisulam. **b** Survival of mice treated with vehicle ($n = 6$ mice) or indisulam ($n = 6$ mice). One mouse in indisulam treated group died on day 7. The autopsy revealed no presence of tumour but evidence of bowel obstruction. **c** Representative anatomical coronal T$_2$-weighted MRI of the abdomen of Th-*MYCN* mice prior (day 0) and following seven days of treatment with indisulam or vehicle control. A dashed white line indicates the tumour circumference. Source data is provided as a Source Data File.

transfection RNA and protein were harvested for downstream analysis of mis-splicing and protein levels.

**Generation of CRISPR–Cas9 DCAF15 knockout KELLY cell lines**. KELLY cells were transduced with Edit-R CRISPR-Cas9-TurboGFP lentiviral particles (Dharmacon #VCAS11864) in a 96 well plate, expanded and GFP positive cells sorted via FACS (total of two rounds). KELLY-Cas9 cells were then transfected with Lipofectamine 2000 (Life Technologies) and either Edit-R crRNA Non-Targeting Control #1 (NTC, Dharmacon, #U-007501-01) or a cocktail of three DCAF15 targeting crRNAs (DCAF15 crRNA 1 5′-GGAGACCCAGAAGAACGGGC-3′, DCAF15 crRNA 2 5′-GCAGCTTCCGGAAGAGGCGA-3′, DCAF15 crRNA 3 5′-ACAGCAAGCTCAAGCTG-3′) and Edit-R CRISPR–Cas9 synthetic tracrRNA (Dharmacon #U-002005). Single-cell clones were expanded for NTC and 3× DCAF15 crRNA conditions and sequenced for DCAF15 disruption.

**Sulforhodamine B (SRB) growth assay**. Cells were seeded overnight in triplicate in 96 well plates and treated with vehicle control (0.1% DMSO) or indisulam or E7820 (Sigma Aldrich, UK) at various doses for 72 h. SRB assay was performed according to the manufacturer's instructions. Data were normalised to vehicle controls.

**Cell viability assays**. Cells were seeded overnight in triplicate in 96-well plates and treated with indisulam, E7820 or vehicle control. Cell viability was established at 72 h after treatment using CellTiter-Glo Luminescent Cell Viability Assay (G7571; Promega) and read on a Synergy HT Multi-Mode Microplate Reader (Biotek). SF50 values were calculated with PRISM GraphPad, and SF50 was defined as the compound concentration required to elicit a 50% inhibition of the cell population compared with the vehicle control.

**Caspase-3/7 assays**. Cells were seeded in a white 96-well plate overnight. The following day, cells were treated with indisulam, E7820 or vehicle control. Forty-eight hours post-treatment, Caspase-3/7 Glo reagent (Promega, UK) was added to the wells in 1:1 (v:v) media. The plate was shaken at 300 rpm for 30 s to induce cell lysis and incubated at room temperature for 1 h. Luminescence was recorded using

a CLARIOstar plate reader (BMG Labtech). Data were analysed on MARS software (v3.40) and first normalised to cell mass (SRB) secondly to vehicle controls.

**Western blotting**. Whole-cell lysates were harvested using RIPA buffer with 1% HALT protease inhibitor (Sigma, UK) and quantified using the BCA protein assay (ThermoScientific Pierce) as per the manufacturer's protocol. Cell lysates containing equal protein concentrations were loaded onto a 4–20% Mini-Protean TGX pre-cast gel (Bio-Rad), before being transferred onto PVDF membranes. Membranes were probed for anti-RBM39 (HPA001591, Sigma), anti-beta-actin (ab8226, Abcam), anti-CDK4 (ab108357, Abcam), anti-TYMS (ab108995, Abcam), anti-n-Myc (OP13, Merckmillipore) and Anti-GAPDH (2118, Cell Signalling) overnight. The following day, membranes were incubated with secondary antibodies for 1 h and chemiluminescent signals were detected. Densitometry analysis of bands was done using Image J (v1.52).

**RNA extraction and PCR**. Total RNA was extracted using the RNeasy Mini kit (Qiagen) according to the manufacturer's protocol. On-column DNase treatment was performed to remove contaminating genomic DNA. RNA concentration and purity were determined using the Nano-Drop Spectrometer (Nano-Drop Technologies, USA). Reverse-transcription (RT) was performed on 2 µg of RNA using the High-Capacity-RNA to cDNA kit (Applied Biosystems, U.K.) in a 20 µL reaction. Exon skipping in *TRIM27*, *CDK4* and *EZH2* was assessed using end-point PCR. Totally, 100 ng cDNA was amplified with Q5® Hot Start High-Fidelity DNA Polymerase (New England BioLabs). Primer details are noted in Table 1.

**Immunofluorescence on tissue sections**. Tumours were processed using an ASP300S tissue processor (Leica) according to the manufacturer's instructions. Sections were de-paraffinized and rehydrated through Histo-Clear and graded alcohol series, rinsed for 5 min in tap water, boiled for 5 min in 1% citric-acid buffer and left to cool to RT. Endogenous enzyme activity was blocked by 1% H$_2$O$_2$ for 20 min followed by 3 washes in ddH$_2$O. Sections were blocked for 1 h in TBS 0.01% Triton (TBST), 5% BSA, RBM39 (HPA001591, Sigma, UK) or Ki67 (Cat #556003, BD Bioscience, UK) antibody was incubated at RT overnight 1:100, washed in TBST, 2nd Alexa Fluor 488/555 goat anti-rabbit antibody was incubated at RT for 1 h (1:500). Antibody solution; TBST, 5% BSA. Images were captured by

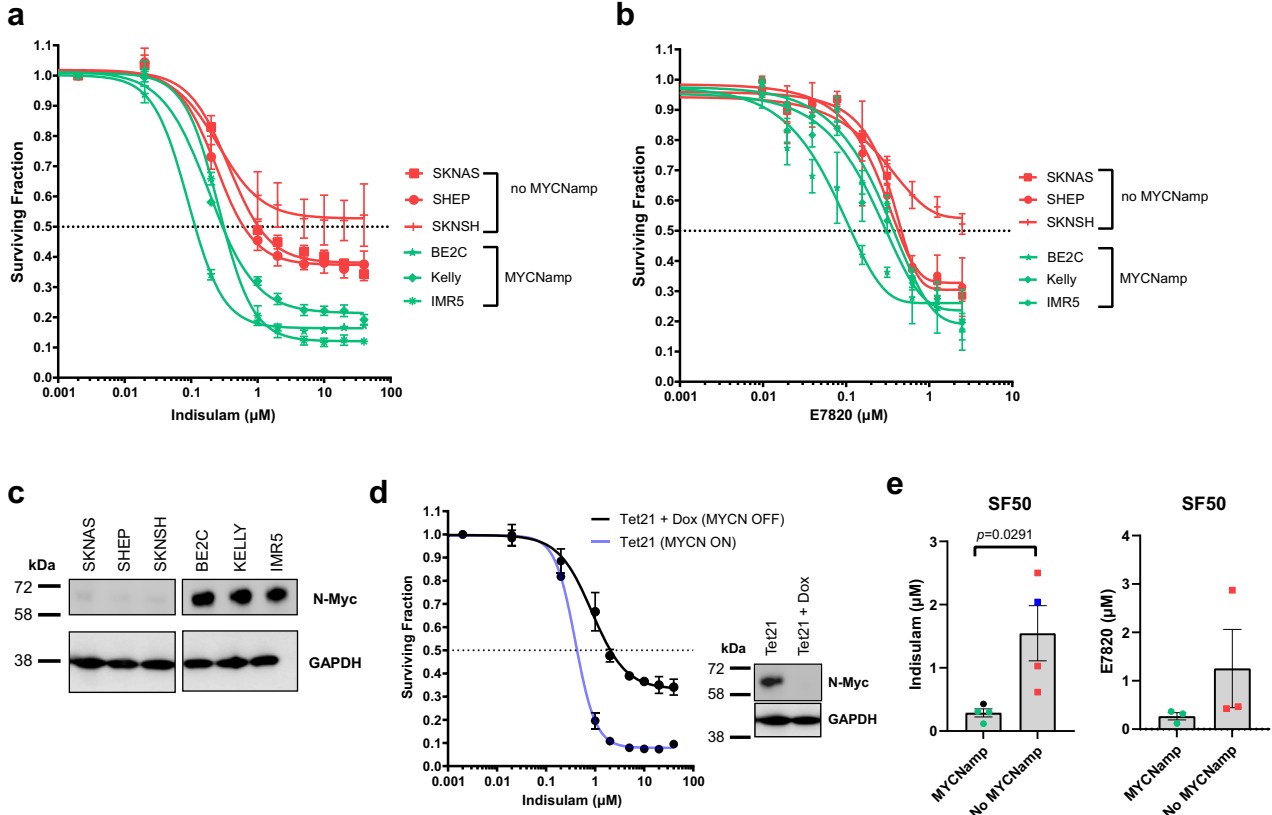

**Fig. 9 *MYCN* expression is a determinant of indisulam sensitivity in neuroblastoma. a**, **b** Dose–response curve in six neuroblastoma lines, three without and three with *MYCN* amplification of indisulam (**a**) or E7820 (**b**). Cell survival was determined after 72 h with Cell-Titre Glo (*n* = 3 independent experiments). **c** Western blot for N-Myc and GAPDH on six neuroblastoma lines. **d** Dose–response curve in Tet21 *MYCN* ON or *MYCN* OFF (treated with 1 µg ml⁻¹ doxycycline) of indisulam. Cell survival was determined after 72 h with Cell-Titre Glo (*n* = 3 independent experiments). Western blot of n-Myc and GAPDH. Blot is representative of *n* = 3 independent experiments. **e** 50% Surviving Fraction (SF50) of neuroblastoma cells (dots represent mean SF50 of *n* = 3 independent experiments) treated with indisulam (*n* = 8 cell lines, 3 *MYCN*amp, 3 no *MYCN*amp, 2 Tet21) or E7820 (*n* = 6 cell lines, 3 *MYCN*amp, 3 no *MYCN*amp). Data are presented as mean ± SD (**a**, **b**, **d**) or SEM (**e**). Statistical analysis in **e** was determined by a two-tailed unpaired *t*-test. Source data is provided as a Source Data File.

**Table 1 PCR primer sequences for end-point PCR.**

| Gene | Forward and reverse sequences 5′–3′ |
|---|---|
| TRIM27 | CCTGAACCTTGGATCACACC |
|  | GCAGGTCCTGTTGGAGGTAA |
| EZH2 | CCGCTGAGGATGTGGATACT |
|  | ACTCTCGGACAGCCAGGTAG |
| CDK4 | GTGTATGGGGCCGTAGGAAC |
|  | CCAACACTCCACATGTCCAC |
| GAPDH | GGCTGCTTTTAACTCTGG |
|  | GGAGGGATCTCGCTCC |

confocal microscopy (LSM700, LSM T-PMT) and processed by ZEN2009 (Black version, Zeiss) software.

**Respiration analysis.** To measure OCR, cells were seeded into a 96 well plate overnight and dosed with indisulam, E7820 or vehicle control for 2 h before the MitoXpress Xtra Oxygen Consumption reagent (Agilent Technologies) and carbonyl cyanide-4 (trifluoromethoxy) phenylhydrazone (FCCP; 2.5 mM) were added. The wells were sealed with warm mineral oil and immediately read using the CLARIOstar plate reader (BMG Labtech) in continuous cycles for 3 h. Fluorescent lifetime (µs) was calculated using the CLARIOstar MARS data analysis software and OCR (µs, hour) was determined by calculating the maximum slope according to manufacturer's instructions and pre-set analysis protocols from BMG Labtech.

**Stable isotope labelling experiments by gas chromatography-mass spectrometry (GC-MS).** For analysing the aqueous (polar) metabolites of KELLY PAR

(parental), DCAF15 WT and DCAF15 KO cells were seeded in a 6-well plate format and left for 24 h to settle to reach ~80% confluency. Cells were dosed with 10 µM indisulam for 24 h in the presence of ¹³C₅-glutamine (2 mM) or ¹³C₆-glucose (5.6 mM) respectively. Wells were washed with Ringer's buffer solution and quenched with 1.5 mL of ice-cold MeOH (Sigma). The resulting solution was dried under a continuous flow of N2. Samples were dual-phase extracted by adding 300 µL ice-cold MeOH:CHCl₃ (1:1) (Sigma), vials were mixed and a further 300 µL ice-cold H₂O (Thermo Fisher) was added. Samples were mixed and centrifuged. The aqueous upper layer was collected in a sylanised glass vial. The dual-phase extraction was repeated once and the resulting supernatant was pooled together. As an internal standard, 10 µL of myristic-acid d27 was added to the vials and samples were snap-frozen and dried (SP Scientific). This was followed by the addition of 20 µL of 2% methoxyamine-hydrogen chloride in pyridine (MOX, Thermo Scientific) and incubated on a heat block (30 °C, 1.5 h). After equilibration at room temperature, 80 µL of N-tert-butyldimethylsilyl-N-methyltrifluoroacetamide (MTBSTFA, Thermo Scientific) with 1% tert-butyldimethylchlorosilane (TBDMS, Thermo Scientific) was added and incubated for 1 h at 70 °C.

In-house protocols adapted from the previous reports[41] were used for GC–MS analysis, a summary is provided here. Totally, 10 µl of 1.5 mg ml⁻¹ myristic acid-d27 (internal standard) was added to each dried extract. Metabolites extracts were derivatised using the two-step method of derivatisation: methoxyamination and silylation[42]. When required, samples were diluted with anhydrous pyridine. Samples were analysed in a splitless mode on an Agilent 7890 GC with a 30 m DB-5MS capillary column and a 10 m Duraguard column. This GC was coupled to an Agilent 5975 MSD[42]. Metabolites were assigned using FiehnLib assisted processing in AMDIS[43] and manually assessed using the Gavin package[41].

**Metabolome analysis by LC–MS/MS**
*In vivo xenograft tumour sample preparation.* Tumour tissues were analysed according to their wet weight and lysed in the presence of glass beads (0.1 mm, Bertin technologies) with 800 µL of pre-chilled 80% MeOH (LC–MS grade, Sigma-

Aldrich) using a Precellys bead beater (Bertin Technologies) at 18,000$g$ for a $4 \times 20$ s cycle with a 30 s pause between each cycle at 4 °C. Samples were then centrifuged 21,000$g$ for 10 min (4 °C, Eppendorf). The supernatant was collected, and the extraction procedure was repeated. A process blank was created as well. All samples and blanks were snap-frozen in liquid $N_2$ and freeze-dried overnight using a freeze-drier (SP Scientific). The samples were then subjected to dual-phase extraction which was previously described. Samples were then diluted 1:18 w/v in 90% ACN (Sigma-Aldrich), based on their wet weight with 90% ACN. From the diluted samples, 50 µL was transferred to a new LC–MS vial and 5 µL of an internal standard mixture (2 mg mL$^{-1}$ for each metabolite) was added to each sample. For quantification, standard curves were prepared alongside samples.

*In vitro cellular sample preparation.* KELLY Parental (PAR), DCAF15$^{WT}$ and DCAF15$^{KO}$ cells were seeded in 6 well plates and allowed to attach and reach 80% confluency. Cells were then treated with fresh media with 10 µM indisulam or 0.1% DMSO (vehicle control). After 24 h incubation, media was carefully removed, and wells were washed with phosphate saline buffer. Wells were quenched with 1.5 mL of pre-chilled 100% MeOH (LC–MS grade, Honeywell), 10 µL of an internal standard mixture (5–50 µg mL$^{-1}$ of stable isotope-labelled modifications of 21 analytes, representative of all metabolite classes) was added to each well. All wells were scrapped, and cellular content was dried down completely under $N_2$ flow. Then, samples were dual-phase extracted as previously described and the polar phase was dried down completely under $N_2$ flow and stored at −80 °C. Upon measurement, samples were thawed on ice and resuspended with 100 µL $H_2O$ and measured by LC–MS/MS in positive and negative modes.

*Metabolite measurements by LC–MS/MS.* Both xenograft tissues and cultured cells were analysed using a QTRAP4000 triple quadrupole mass spectrometer (AB Sciex, Dahaner Corporation) coupled to a 1290 Infinity UPLC system (Agilent Technologies). A 15 min normal-phase chromatographic method was used for both positive and negative ionisation modes, using an Acquity UPLC BEH Amide column (Waters Corporation). Mobile phases were prepared with UPLC grade solvents and chemicals as follows: 100% water with 0.1% formic acid and 20 mM ammonium formate (positive mode aqueous), 100% acetonitrile with 0.1% formic acid (Positive mode organic), 100% water with 10 mM ammonium hydroxide and 20 mM ammonium acetate (Negative mode aqueous) and 100% acetonitrile with 10 mM ammonium hydroxide (Negative mode organic). Multiple reaction monitoring transitions were used for all analytes and internal standards, validated with pure chemical standards (Sigma). Quality control pooled samples were used in all experiments, and metabolites out of a QC accuracy range of 70–130% were removed from the produced datasets.

Statistical analysis was performed using MetaboAnalyst[44] (v5.0). The blank-corrected and QC-checked relative intensities were normalised to the total sum of metabolites in the sample, log-transformed, and auto-scaled (mean-centred and divided by the standard deviation of each variable). Kruskal–Wallis non-parametric analysis was performed, and the resulting most significant metabolites ($p < 0.001$) were included in a heatmap with metabolite hierarchical clustering using Ward's minimum variance method.

## Protein identification and quantification by LC–MS/MS

*Sample processing.* Protein samples (50 µg/replicate) were processed using the Filter Aided Sample Preparation protocol[45]. Briefly, samples were loaded onto 30 kDa centrifugal concentrators (Millipore, MRCF0R030) and buffer exchange was carried out by centrifugation on a benchtop centrifuge (15 min, 12,000$g$). Multiple buffer exchanges were performed sequentially with UA buffer (8 M urea in 100 mM Tris pH 8.5, $3 \times 200$ µl), reduction with 10 mM DTT in UA buffer (30 min, 40 °C) and alkylation with 50 mM chloroacetamide in UA buffer (20 min, 25 °C). This was followed by buffer exchange into UA buffer ($3 \times 100$ µl) and 50 mM ammonium bicarbonate ($3 \times 100$ µl). Digestion was carried out with mass spectrometry grade trypsin (Promega, V5280) using 1 µg protease per digest (16 h, 37 °C). Tryptic peptides were collected by centrifugation into a fresh collection tube (10 min, 12,000$g$) and washing of the concentrator with 0.5 M sodium chloride (50 µl, 10 min, 12,000$g$) for maximal recovery. The following acidification with 1% trifluoroacetic acid (TFA) to a final concentration of 0.2%, collected protein digests were desalted using Glygen C18 spin tips (Glygen Corp, TT2C18.96) and peptides eluted with 60% acetonitrile, 0.1% formic acid (FA). Eluents were then dried using vacuum centrifugation.

*LC–MS/MS peptide analysis.* Dried tryptic digests were re-dissolved in 0.1% TFA by shaking (1200 rpm) for 30 min and sonication on an ultrasonic water bath for 10 min, followed by centrifugation (20,000$g$, 5 °C) for 10 min. LC–MS/MS analysis was carried out in technical duplicates and separation was performed using an Ultimate 3000 RSLC nano liquid chromatography system (Thermo Scientific) coupled to a Q-Exactive mass spectrometer (Thermo Scientific) via an EASY spray source (Thermo Scientific). For LC–MS/MS analysis protein digest solutions were injected and loaded onto a trap column (Acclaim PepMap 100 C18, 100 µm × 2 cm) for desalting and concentration at 8 µl/min in 2% acetonitrile, 0.1% TFA. Peptides were then eluted online to an analytical column (Acclaim Pepmap RSLC C18, 75 µm × 75 cm) at a flow rate of 200 nl/min. Peptides were separated using a 120 min gradient, 4–25% of buffer B for 90 min followed by 25–45% buffer B for

another 30 min (composition of buffer B—80% acetonitrile, 0.1% FA) and subsequent column conditioning and equilibration. Eluted peptides were analysed by the mass spectrometer operating in positive polarity using a data-dependent acquisition mode. Ions for fragmentation were determined from an initial MS1 survey scan at 70,000 resolution, followed by higher energy collision-induced dissociation of the top 12 most abundant ions at 17,500 resolution. MS1 and MS2 scan AGC targets were set to 3e6 and 5e4 for maximum injection times of 50 ms and 50 ms respectively. A survey scan m/z range of 400–1800 was used, normalised collision energy set to 27%, charge exclusion enabled with unassigned and +1 charge states rejected and a minimal AGC target of 1e3.

*Raw data processing.* Data were processed using the MaxQuant software platform (v1.6.1.0), with database searches carried out by the in-built Andromeda search engine against the Uniprot *H. sapiens* database (version 20180104, number of entries: 161,521). A reverse decoy search approach was used at a 1% false-discovery rate for both peptide spectrum matches and protein groups. Search parameters included: maximum missed cleavages set to 2, fixed modification of cysteine carbamidomethylation and variable modifications of methionine oxidation, protein N-terminal acetylation and serine, threonine, tyrosine phosphorylation. Label-free quantification was enabled with an LFQ minimum ratio count of 2. 'Match between runs' function was used with match and alignment time limits of 1 and 20 min, respectively.

**Genomic analysis.** Total RNA was extracted using the RNeasy Mini kit (Qiagen) and quantified by Qubit (Thermo Fisher) to assess sample integrity. Ribosomal and mitochondrial RNA was removed via ribodepletion. Seventy-five base paired-end reads were sequenced by the Imperial BRC Genomics Facility using the HiSeq4000 (Illumina) resulting in ~45 million reads per sample. Data were aligned to the human reference genome (version hg19) using HISAT2 (v 2.1.0) and BAM files were visualised using the Integrative Genome Viewer (Broad Institute). Read counts were quantified using function feature Counts from the R package Rsubread (v 1.34.7).

Proteomics analysis from IMR-32 cells subjected to 5 µM indisulam for 6 and 16 h was used for further analysis. The mean of three duplicates was taken and fold changes were calculated relative to the corresponding vehicle control. *P*-values were corrected using the Benjamini–Hochberg method and were considered significant if <0.05. Further, genes that were identified as mis-spliced were compared with statistically significant up and downregulated proteins (at 16 h indisulam exposure versus vehicle control). The overlap of which was then imported into the Consensus PathDB interaction database (provided by Max Planck Institute for Molecular Genetics http://cpdb.molgen.mpg.de/) for pathway enrichment analysis.

**Alternative splicing analysis.** SpliceFisher (github.com/jiwoongbio/SpliceFisher) was used to detect alternative splicing events, where exon and intron regions were defined from the hg19 human reference genome. To estimate differential exon skipping events, the number of exon-junction reads and exon skipping reads were calculated and compared with the vehicle control (Supplementary Fig. 1). Alternatively, for differential intron retention, the numbers of exon-intron and exon-exon reads were used. Read counts were evaluated by multiple Fisher's exact tests in three two-by-two tables using R, and p-values were adjusted using the Benjamini–Hochberg method and deemed significant $p < 0.05$.

**In vivo experiments.** All experimental protocols were approved and monitored by The Institute of Cancer Research Animal Welfare and Ethical Review Body (PPL no. 70/7945, later PPL P91E52C32), in compliance with the UK Home Office Animals (Scientific Procedures) Act 1986, the United Kingdom National Cancer Research Institute guidelines for the welfare of animals in cancer research[46] and the ARRIVE guidelines[47].

*In vivo tumour xenografts of IMR-32 neuroblastoma cells.* NCr-Foxn1$^{nu}$ mice (female, age 6 weeks, weight 24.6 g ± 1.4) were injected subcutaneously unilaterally with $1.5 \times 10^6$ IMR-32 cells with 30% matrigel (100 µl total). Callipers were used to measure tumour diameter on two orthogonal axes 2–3 times per week. Volume was calculated using the equation; $v = 4/3\pi r^3$ (where $r =$ radius, calculated as an average of the two axes). Dosing occurred at a predetermined mean tumour size of 0.24 cm$^3$ ± 0.09 or 7 mm ± 0.09 mm for 8 continuous days, *intravenous* with either indisulam 25 mg kg$^{-1}$ or vehicle only (3.5% DMSO and 6.5% Tween 80 in saline). This dose was chosen guided by other studies[6,32] and based on guidance from Nair & Jacob[48] this equates to a dose of ~105 mg m$^{-2}$, which compares favourably to reported tolerated doses in adult humans of 500–700 mg m$^{-2}$ [49]. Survival studies were terminated when the mean diameter of the tumour reached 1.9 cm$^3$ or after 66 days of the tumour-free disease, vehicle ($n = 6$), indisulam ($n = 6$). (A tumour of 1.9 cm$^3$ is the maximal size permitted by the Institute of Cancer Research Animal Welfare and Ethical Review Body was not exceeded). PD study; mice were treated for four days. Tumour tissues were harvested (snap frozen or fixed in 4% paraformaldehyde) for further analysis, vehicle ($n = 5$), indisulam ($n = 5$).

*In vivo transgenic model of neuroblastoma.* Th-*MYCN* mice (129×1/SvJ-Tg(Th-*MYCN*)41Waw/Nci, male and female, age 71 days ± 12 days, weight 24.3 g ± 3.2)

have been described previously[27]. In this study, we used heterozygous Th-*MYCN* mice in which we observe a 30% penetrance with tumour onset of $73 \pm 25$ days. Two to five mice were caged together and were allowed access to sterilised food and water *ad libitum*. Th-*MYCN* mice were monitored by palpation twice weekly and $n = 12$ mice were randomly enrolled into the control group ($n = 6$) or treated group ($n = 6$) when tumours reached a palpation score of approximately 5 mm in diameter. Dosing was done for 8 continuous days, *intravenous* with either indisulam 25 mg kg$^{-1}$ or vehicle only (3.5% DMSO and 6.5% Tween 80 in saline). Studies were terminated when the tumour grew to a palpable size of 10 mm or immediately upon showing any signs of ill health. A tumour of 10 mm is the maximal size permitted by the Institute of Cancer Research Animal Welfare and Ethical Review Body was not exceeded. For survival analysis, mice were treated with indisulam were monitored for up to 124 days ($n = 5$). Tumour size, animal weight, and overall animal well-being were scored daily throughout the study. One animal from the treated group was culled due to generalised ill-health at day 7; autopsy showed no macroscopic tumour residual but evidence of bowel obstruction.

MR images were acquired on a 1 Tesla M3 small animal MRI scanner (Aspect Imaging, Shoham, Israel). Mice were anaesthetised using isoflurane delivered via oxygen gas and their core temperature was maintained at 37 °C. Anatomical fat-suppressed T$_2$-weighted coronal images (TE = 9 ms, TR = 4600 ms) were acquired from 20 contiguous 1-mm-thick slices through the mouse abdomen, from which tumour volumes were determined using segmentation from regions of interest drawn on each tumour-containing slice using Horos medical image viewer (v3.3.6). MRI was performed on a subgroup of mice throughout the study subject to equipment availability ($n = 4$ vehicle, $n = 3$ indisulam).

**Statistics and reproducibility**. Data were analysed in GraphPad Prism (v9.2.0) and R. Data shown from representative experiments were conducted with similar results. Source data are provided with this paper.

**Reporting summary**. Further information on research design is available in the Nature Research Reporting Summary linked to this article.

## Data availability

The mass spectrometry proteomics data generated in this study have been deposited to the ProteomeXchange Consortium via the PRIDE[50] partner repository under the accession code PXD022164. The RNAseq data generated in this study have been deposited in NCBI's Gene Expression Omnibus[51] under the accession number GSE160446. Processed metabolome data generated in this study are available as Supplementary Data files. CERES CRISPR and CTRP data were downloaded from DepMap database (https://depmap.org/portal/). *DCAF15* expression data from cancer cell lineages were downloaded from CCLE RNA-seq (https://portals.broadinstitute.org/ccle). Source data are provided with this paper.

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

## Acknowledgements

This work was funded by the Imperial College Medical Research Council (MRC) Doctoral Training Programme (DTP), grant number: MR/K501281 and by MetaCELL-TM H2020 Fast Track to Innovation (737978). Infrastructure for this research was supported by the Imperial Experimental Cancer Medicine Centre and the Cancer Research UK Imperial Centre as well as the National Institute for Health Research (NIHR) Imperial Biomedical Research Centre (including the Imperial Genomics Facility & metabolomics support by Dr. Alexandros Siskos and Eirini Kouloura). Cancer Research UK also support the Cancer Imaging Centre at ICR, in association with the MRC and Department of Health (England) (C1060/A16464). O.Y., E.P., B.M.C., and L.C. are funded by Cancer Research UK (C34648/A18339 and C34648/A28278). Y.J. is a Children with Cancer UK Research Fellow (2014/176). A.N. is funded through an AstraZeneca/NIHR Imperial BRC Imperial College Research Fellowship (2019). We acknowledge and thank Flora Upton for support with the generation of figures.

## Author contributions

A.N., A.S., O.Y., Y.J., L.C. and H.C.K., conceived the study, designed the experiments and interpreted data. A.N., A.S., L.H., E.P., Y.L., Y.M. and Y.X acquired data from cellular experiments with the guidance of A.B. and E.W. on methodology. M.G., Y.M., C.W. and A.N generated CRISPR–Cas9 DCAF15 clones and conducted experiments. O.Y., Y.J. and B.M.C. performed and acquired data from in vivo experiments, including imaging. H.K. and A.M. performed proteomics and processing of data with the guidance of D.C. on methodology. A.S., L.H. and C.B.. performed mass spectrometry of in vitro and in vivo samples, processed and analysed data. C.E. and G.V. analysed RNAseq and proteomics data. All authors contributed to the writing and editing of the paper.

## Competing interests

The authors declare no competing interests.
