## [Peer Review File · Nature Communications]

Indisulam targets RNA splicing and metabolism to serve as a therapeutic strategy for high-risk neuroblastomaREVIEWER COMMENTS

Reviewer #1 (Remarks to the Author):

This manuscript presents data revealing the sensitivity of neuroblastoma to RBM39 degradation by indisulam. While the data shown are all well performed, there is very little shown that is unexpected regarding the mechanism of action of indisulam based on prior studies or the basis for an effect of this compound in this specific disease (even most of the mis-splicing targets of indisulam are from previously published data). Here are some key questions that would be important to address to raise the level of novelty in this study:

- What is the basis for the preferential sensitivity of neuroblastoma to RBM39 degradation over other cancer types? Is it related to DCAF15 mRNA expression, MYC activation, something else? There are multiple molecular subtypes of neuroblastoma—are all similarly sensitive to RBM39 degradation?
- The authors should evaluate some subset of these experiments using an RBM39 degrading compound besides indisulam (e.g. E7820, CQS, etc). The current manuscript relies solely on one drug.
- Indisulam and related compounds also degrade RBM23. What is the role of RBM23 in the neuroblastoma models studied here? Is RBM23 required for their cell survival?
- The data cited about older studies of indisulam on carbonic anhydrase IX have been questioned based on the recent observations by multiple groups that loss of DCAF15 or specific RBM39 mutations can rescue all of the effects of indisulam. The manuscript does not reflect this current knowledge appropriately.

Reviewer #2 (Remarks to the Author):

Neuroblastoma is a very rare type of pediatric cancer characterized by a heterogeneous clinical phenotype and a poor prognosis, especially in patient subgroup with MYCN amplification. In the manuscript entitled "Indisulam targets RNA splicing and metabolism to serve as a novel therapeutic strategy for high-risk neuroblastoma" Nijhuis and co-authors propose to use indisulam (E7070), a synthetic sulfonamide cell cycle inhibitor previously tested in preclinical and clinical studies on multiple cancers both as a single treatment or in combination with other chemotherapeutic agents.

Although multiple clinical trials showed poor results of E7070 treatment in patients diagnosed with solid and blood cancers (e.g. melanoma, leukemia, and squamous cell carcinoma of the head and neck), the investigators show that E7070 could be a promising treatment in high-risk neuroblastoma patients. To prove the efficacy of indisulam treatment, the investigators capitalized on multiple biological and analytical techniques and tested the agent in in vitro and in mouse models showing that E7070 abrogates proteins involved in cell cycle and metabolism and targets splicing by inducing RBM39 degradation via recruitment to DCAF15.

The use of patient-derived xenografts as preclinical neuroblastoma models would be more relevant to suggest E7070 treatment.

In most of the experiments, the investigators use acute treatment for a short time (10uM, 6 hours) that seems to target several metabolic pathways. Metabolic perturbations following treatment are in general very mild. In fact, if the authors would correct the p-values using false discovery rate, as they should, not many metabolites would be statistically significant in both in vitro and in vivo experiments. Increasing the number of mice could help the investigators to detect more significant metabolic changes induced by treatment.

Metabolic flux analysis using ¹³C isotopic tracers is encouraged to understand the metabolic changes inside and outside the mitochondria (e.g. glucose versus glutamine metabolism) and these might be more significant compared to the levels of the metabolic data.

Consideration about redox factors activity (line ~ 416) is too speculative and it should be revised or removed. Many reactions can contribute to the NADH/NAD and FADH₂/FAD changes. The authors do not report FADH₂/FAD measurements, and they did not perform fatty acid and lipidomics analyses, so any assumption about FADH₂ and FAD is very weak.

Reviewer #3 (Remarks to the Author):

This is a clearly written paper describing the mechanism of action and efficacy of indisulam in the treatment of neuroblastoma. Starting with global data, they identify that neuroblastoma is sensitive to indisulam from published AUC data and therefore investigate this further using both cell lines and murine models. Initially, they focus on the mechanism of action and perform elegant experiments showing that indisulam targets the RNA splicing factor RBM39 for proteosomal degradation via the DCAF15 E3 ubiquitin ligase resulting in splicing errors ultimately affecting cell growth. Furthermore, they show using mouse models that indisulam is effective in preventing tumour growth and therefore is a potential therapeutic option for NB patients.

Overall, this is a nice study that clearly shows the mechanism of action of indisulam as it applies to neuroblastoma and shows its efficacy in cell lines and murine models. My main issue is that the authors have not explored why some NB are more susceptible to indisulam than others. I think this is an important avenue to investigate as it has obvious clinical implications.

Major comments:

1. In figure 1, the IC50 values are given as ranging from ~8-13nM for each of the cell lines assessed yet in Figure 1d, the drug concentration required to decrease cell viability is much higher ranging from 1000-10000nM - can the authors account for this discrepancy? It is not clear to me if the IC50 values are true IC50 or ED50s? I suspect the latter as the output is cell viability. Additionally, the IC50 for LS and SHEP cell lines do not differ significantly from the other 2 cell lines yet these 2 cell lines are far more resistant to the effects of Indisulam - again, can the authors clarify why this is the case?
2. Are the n values given in Figure 1 technical or biological replicates? This should be stated here as well as throughout the manuscript. There is no indication in most places how many replicates were performed. Are the WB representative of biological replicates for example? The RT-PCR data?
3. It is obvious why subsequent experiments are conducted with IMR-32 and KELLY cell lines but could the authors provide an explanation as to the relative lack of response for the other 2 cell lines? Is this due to their MYCN status? ALK? ATRX? etc... This is important knowledge to perhaps distinguish which patients might best benefit from indisulam.
4. In the mouse models, the mice are dosed when the tumours are relatively small, not really mimicking the human scenario. How do the mice respond if the tumours are larger at the commencement of treatment (although I appreciate that animal welfare often precludes such studies)? The authors also employ 1 cell line xenograft and 1 GEMM, therefore raising the question as to whether the effects they observe will be broadly applicable to all NB patients. Referring to point 3 above, are there specific genetic differences in individual patient tumours that would indicate a patient's potential sensitivity to indisulam? The use of patient derived xenografts (with a broader range of genetic backgrounds) might help to elucidate this. In lieu of this, the sensitivity of a broader range of cell lines would help to delineate sensitivity factors.

Minor comments:

1. Abstract line 29, a word is missing in the sentence: complete tumour xxxxx without relapse....
2. Figure 2C - VC data are missing - it is important to show the background rate for comparison.
3. Figure 4c is rather subjective for CDK4, particularly for the KELLY cell line. If biological replicates were conducted, it should be simple to determine densities of the bands for the blots to provide statistical data.
4. Can the authors comment on how the in vivo dosage of 35mg/kg was arrived at and whether this dose is relevant to the human situation?

Reviewer #1

1. What is the basis for the preferential sensitivity of neuroblastoma to RBM39 degradation over other cancer types? Is it related to DCAF15 mRNA expression, MYC activation, something else? There are multiple molecular subtypes of neuroblastoma—are all similarly sensitive to RBM39 degradation

DCAF15 is the main determinant for sensitivity when cell lines from the CCLE are analysed. mRNA expression is strongly correlated to indisulam AUC in public cell line data bases analysed by us (Supplementary Figure S6) and others (Han et al 2017). Neuroblastoma lines have the highest *DCAF15* expression profile among lines derived from solid tumours (Supplementary Figure S6). As *MYCN* amplification results in an oncogenic transcription that requires functional RNA splicing, we also tested the hypothesis that aryl sulphonamides could cause preferential toxicity in *MYCN*-amp cell lines. We have now added dose-response curves of indisulam and E7820 in additional neuroblastoma cell lines with varying levels of *MYCN*-amplification (three non-*MYCN*-amp, three *MYCN*-amp) and one N-myc inducible cell line (Tet21 dox model) showing that *MYCN* is indeed a determinant factor for indisulam sensitivity in neuroblastoma (Fig. 9).

2. The authors should evaluate some subset of these experiments using an RBM39 degrading compound besides indisulam (e.g. E7820, CQS, etc). The current manuscript relies solely on one drug

We agree with the reviewer's comment and have included E7820 as a second RBM39-degrading compound in some of our experiments. We show that neuroblastoma cell lines are also sensitive to E7820 in a *DCAF15*-dependent manner (Fig. 4)

3. Indisulam and related compounds also degrade RBM23. What is the role of RBM23 in the neuroblastoma models studied here? Is RBM23 required for their cell survival?

Although RBM23 degradation by indisulam was detected in other proteomic studies (Ting et al 2019, Bussiere et al 2020), it was below the detection limit in our analysis in IMR-32 cells. We did not further investigate RBM23 in our models for the following reasons:

1. The loss of RBM23 does not result in RNA splicing changes as observed with loss of RBM39 (Ting 2019).
2. Mutations in the RBM39-indisulam binding site rescues sensitivity to indisulam in HCT116 (Han et al., 2017).

4. The data cited about older studies of indisulam on carbonic anhydrase IX have been questioned based on the recent observations by multiple groups that loss of DCAF15 or specific RBM39 mutations can rescue all of the effects of indisulam. The manuscript does not reflect this current knowledge appropriately.

To address this we generated *DCAF15*-WT and KO cell lines. Indeed, *DCAF15* loss rescues to majority of cytotoxic effects of both indisulam and E7820 in neuroblastoma. At higher doses (10-20 μ M or 1-2 orders of magnitude above the ED50 dose) we still see a significant reduction in cell growth (20-40%) which could be contributed to two things:

- 1) Residual functional *DCAF15* expression in a subclone. The lack of a commercially available and well-validated *DCAF15* antibody means we are unable to fully test this hypothesis.
- 2) Some indisulam-mediated metabolic changes that we observed were *DCAF15*-independent and thus RBM39-independent (Figures 6 & 7). These could be responsible for

some residual toxicity, particularly as *MYCN*amp lines may be more sensitive to metabolic perturbations.

Reviewer #2 (Remarks to the Author):

Major comments:

1. The use of patient-derived xenografts as preclinical neuroblastoma models would be more relevant to suggest E7070 treatment.

While we agree with the reviewer that PDX data would be of value, this is beyond the scope of the current publication in which two different pre-clinical models are already presented.

2. In most of the experiments, the investigators use acute treatment for a short time (10uM, 6 hours) that seems to target several metabolic pathways. Metabolic perturbations following treatment are in general very mild. In fact, if the authors would correct the p-values using false discovery rate, as they should, not many metabolites would be statistically significant in both in vitro and in vivo experiments. Increasing the number of mice could help the investigators to detect more significant metabolic changes induced by treatment.

We have now included new metabolomic data *in vitro* following longer exposure to indisulam (10 μ M, 24 h, Figure 6 and 7) which shows consistent and significant metabolic responses. These exploratory 'discovery' analyses have been adjusted for FDR. As *in vivo* experiments were intended to be confirmatory of the discovery experiments *in vitro* we did not adjust for false discovery for these data. We did not have the opportunity for additional *in vivo* metabolomic experiments.

3. Metabolic flux analysis using ¹³C isotopic tracers is encouraged to understand the metabolic changes inside and outside the mitochondria (e.g. glucose versus glutamine metabolism) and these might be more significant compared to the levels of the metabolic data.

We fully agreed with the reviewer's comment and have conducted ¹³C isotopic tracer experiments using both glutamine and glucose in KELLY DCAF15-WT and KO lines (Figure 6 and 7). We show that indisulam globally impacts on the metabolic flux from both carbon sources, in particular demonstrating that utilisation of glucose carbon into the TCA cycle and in *de novo* lipids synthesis is impaired in a DCAF15/RBM39 dependent manner. interestingly, alterations in metabolic flux from glutamine could be DCAF15-independent. This is an important novel observation and may be a contributing factor to the hyper-sensitivity in neuroblastoma models.

4. Consideration about redox factors activity (line ~ 416) is too speculative and it should be revised or removed. Many reactions can contribute to the NADH/NAD and FADH₂/FAD changes. The authors do not report FADH₂/FAD measurements, and they did not perform fatty acid and lipidomics analyses, so any assumption about FADH₂ and FAD is very weak.

We agree with the reviewer's comment and have removed the relevant statements. We have however now included data on lipid synthesis as described in point 3 above.

Reviewer #3 (Remarks to the Author):

Major comments:

1. In figure 1, the IC50 values are given as ranging from ~8-13nM for each of the cell lines assessed yet in Figure 1d, the drug concentration required to decrease cell viability is much higher ranging from 1000-10000nM - can the authors account for this discrepancy? It is not clear to me if the IC50 values are true IC50 or ED50s? I suspect the latter as the output is cell viability. Additionally, the IC50 for LS and SHEP cell lines do not differ significantly from the other 2 cell lines yet these 2 cell lines are far more resistant to the effects of Indisulam - again, can the authors clarify why this is the case?

There appeared to be a mistake in the IC50 calculation which has been corrected. As also indicated by the reviewer, we have subsequently decided to report ED50 (or SF50) outputs which are more appropriate for the assays conducted.

We have also removed the cell viability data (original submission Figure 1d) as these experiments were conducted in different plate formats/cell numbers which impacts on growth dynamics and subsequent drug efficacy. Now, we have conducted all cell growth assays in 96-well plates for consistency.

2. Are the n values given in Figure 1 technical or biological replicates? This should be stated here as well as throughout the manuscript. There is no indication in most places how many replicates were performed. Are the WB representative of biological replicates for example? The RT-PCR data?

We have included information on experimental replication in every legend.

3. It is obvious why subsequent experiments are conducted with IMR-32 and KELLY cell lines but could the authors provide an explanation as to the relative lack of response for the other 2 cell lines? Is this due to their MYCN status? ALK? ATRX? etc... This is important knowledge to perhaps distinguish which patients might best benefit from indisulam.

The lack of reduced cell viability reported in two cell lines (LS and SHEP) in the original submission (Figure 1d) is likely a technical anomaly as mentioned in point 1 above and thus has been removed from our revised manuscript. Instead, we have expanded our cell line panel and focussed on delineating *MYCN* amplification as a determinant for sensitivity to indisulam and E7820 (Figure 9).

4. In the mouse models, the mice are dosed when the tumours are relatively small, not really mimicking the human scenario. How do the mice respond if the tumours are larger at the commencement of treatment (although I appreciate that animal welfare often precludes such studies)? The authors also employ 1 cell line xenograft and 1 GEMM, therefore raising the question as to whether the effects they observe will be broadly applicable to all NB patients. Referring to point 3 above, are there specific genetic differences in individual patient tumours that would indicate a patient's potential sensitivity to indisulam? The use of patient derived xenografts (with a broader range of genetic backgrounds) might help to elucidate this. In lieu of this, the sensitivity of a broader range of cell lines would help to delineate sensitivity factors.

1. IMR-32 xenograft tumours were grown to a size that is similar to other IMR-32 xenograft studies (200mm³, Subramanian et al., 2016). However, now that we have observed a strong response both *in vivo* models, future studies could explore the impact of indisulam on late stage disease and we will take this recommendation forward.

2. We agree with the reviewer that PDX studies would be valuable. Unfortunately, the use of PDX was outside the scope and timeline of this submission. However, new data in this revised manuscript does show that MYCN status is a determinant of indisulam response across a broader range of neuroblastoma cell lines. Sensitivity was also observed in cells with additional oncogenic drivers such as ALK (KELLY-ALK^{F1174L}).

Minor comments:

1. Abstract line 29, a word is missing in the sentence: complete tumour xxxxx without relapse....

The missing word has been completed. The abstract line now reads: "Complete tumour regression without relapse was observed"

2. Figure 2C - VC data are missing - it is important to show the background rate for comparison.

Given the broad range of alternative splicing variants in different cell types and cancer backgrounds, SpliceFisher is a differential analysis that compares RNA counts proximal to splice sites between treatment and control group. Data shown in the table referred to are the number of RNA splicing events with read counts differential to VC and thus the VC are not shown (by definition would be 0). However, when validating individual mis-splicing events with PCR, VC data are shown.

3. Figure 4c is rather subjective for CDK4, particularly for the KELLY cell line. If biological replicates were conducted, it should be simple to determine densities of the bands for the blots to provide statistical data.

CDK4 protein expression were determined in three biological replicates and densitometry analysis has been added to the figure (Fig. 3). In addition, we have shown that the impact on CDK4 (and TYMS) are dependent on DCAF15 and thus downstream for RBM39 degradation (Fig. 4)

4. Can the authors comment on how the *in vivo* dosage of 35mg/kg was arrived at and whether this dose is relevant to the human situation?

Dose was chosen due to the literature available made by others indicating mice are able to tolerate these doses with little to no side effects. Based on guidance from Nair & Jacob, this equates to a dose of ~105 mg / m² in human, which compares favourably to reported tolerated doses in adult humans of 500-700 mg/m² (e.g. Siegel-Lakhai et al 2008). We have added information in the method section for completeness.

REVIEWERS' COMMENTS

Reviewer #1 (Remarks to the Author):

The authors have adequately responded to my initial comments and questions. I have no further issues with the manuscript.

Reviewer #2 (Remarks to the Author):

The authors have now included in the revised manuscript a substantial amount of new experiments that reinforce their hypothesis about indisulam and its activity in targeting both metabolism and RNA splicing for high-risk neuroblastoma. The addition of MFA data supports previous observations and provides interesting observations about hyper-sensitivity in neuroblastoma models and glutamine metabolism. I recommend the manuscript for publication.

Reviewer #3 (Remarks to the Author):

The authors have sufficiently addressed my concerns.

I still feel that analysis of a PDX model would be beneficial to the manuscript but the addition of cell line data showing the relevance of MYCN amplification for this treatment approach is perhaps a good compromise.